# ON MINIMIZING ADVERSARIAL COUNTERFACTUAL ERROR IN ADVERSARIAL RL

**Roman Belaire**
Singapore Management University
Singapore
`rbelaire.2021@phdcs.smu.edu.sg`

**Arunesh Sinha**
Rutgers University
New Brunswick, NJ
`arunesh.sinha@rutgers.edu`

**Pradeep Varakantham**
Singapore Management University
Singapore
`pradeepv@smu.edu.sg`

## ABSTRACT

Deep Reinforcement Learning (DRL) policies are highly susceptible to adversarial noise in observations, which poses significant risks in safety-critical scenarios. The challenge inherent to adversarial perturbations is that by altering the information observed by the agent, the state becomes only partially observable. Existing approaches address this by either enforcing consistent actions across nearby states or maximizing the worst-case value within adversarially perturbed observations. However, the former suffers from performance degradation when attacks succeed, while the latter tends to be overly conservative, leading to suboptimal performance in benign settings. We hypothesize that these limitations stem from their failing to account for partial observability directly. To this end, we introduce a novel objective called Adversarial Counterfactual Error (ACoE), defined on the beliefs about the true state and balancing value optimization with robustness. To make ACoE scalable in model-free settings, we propose the theoretically-grounded surrogate objective Cumulative-ACoE (C-ACoE). Our empirical evaluations on standard benchmarks (MuJoCo, Atari, and Highway) demonstrate that our method significantly outperforms current state-of-the-art approaches for addressing adversarial RL challenges, offering a promising direction for improving robustness in DRL under adversarial conditions. Our code is available at https://github.com/romanbelaire/acoe-robust-rl.

## 1 INTRODUCTION

The susceptibility of Deep Neural Networks (DNNs) to adversarial attacks on their inputs is a well-documented phenomenon in machine learning (Goodfellow et al., 2014; Madry et al., 2017). Consequently, Deep Reinforcement Learning (DRL) models are also vulnerable to input perturbations, even when the environment remains unchanged (Gleave et al., 2019; Sun et al., 2020; Pattanaik et al., 2017). As DRL becomes increasingly relevant to real-world applications such as self-driving cars, developing robust policies is of paramount importance (Spielberg et al., 2019; Kiran et al., 2021). An example highlighted by Chen et al. (2018) successfully alters a stop sign both digitally and physically to deceive an object recognition model, demonstrating the ease and potential dangers of such adversarial attacks.

Adversarial retraining, which entails inserting adversarial perturbations to the replay buffer during training, effectively enhances the robustness of deep reinforcement learning (DRL) against known adversaries (Gleave et al., 2019; Goodfellow et al., 2014; Pattanaik et al., 2017; Sun et al., 2023). However, this approach often fails to generalize well to out-of-sample adversaries (Gleave et al., 2019; Guo et al., 2023). More importantly, it is well-known that stronger adversaries can always be found (Madry et al., 2017) and that the high-dimensional observation spaces of real problems offer an overwhelming number of adversarial directions (Korkmaz, 2023; Liu et al., 2023). Furthermore,

due to issues such as catastrophic forgetting, naive adversarial retraining in reinforcement learning can result in unstable training processes and diminished agent performance (Zhang et al., 2020). This highlights the need for algorithms that are not tailored to niche adversarial perturbations but are generally robust. Rather than develop a policy that is value-optimal for a set of known adversarial examples, our goal is to identify and mitigate behaviors and states that introduce unnecessary risk. A widely-recognized method to achieve general robustness is the maximin optimization, which seeks to maximize the minimum reward of a policy (Everett et al., 2020; Liang et al., 2022). While this approach does enhance safety, it often sacrifices the quality of the unperturbed solution to improve the worst-case scenario.

Another prevalent robustness mechanism strengthens "non-adversarial value" optimizing policies (i.e. vanilla policies) by incorporating adversarial loss regularization terms, ensuring robust policies are close to the "non-adversarial value" optimal policies. This aims to ensure that actions remain consistent across similar observations, thereby reducing the likelihood of successful adversarial attacks (Oikarinen et al., 2021; Zhang et al., 2020; Liang et al., 2022). However, prior empirical findings indicate that these methods still leave policies vulnerable when attacks do succeed (Belaire et al., 2024), as the observation space is high-dimensional; it is not feasible to ensure *all* similar observations have similar actions.

Adversarial perturbations make the ground truth partially observable and this aspect–though acknowledged–has not been explicitly reasoned within existing work, except recently in Liu et al. (2024); McMahan et al. (2024), the best-performing of which is called Protected (Liu et al., 2024). However, the Protected framework requires multiple adaptation runs at test time to achieve better performance than existing work. The requirement for multiple execution runs in the presence of an adversary at test time is not viable in self-driving cars and other real-world scenarios. To that end, we introduce a novel objective called Adversarial Counterfactual Error (ACoE), which calculates the error due to adversarial perturbations by explicitly considering the belief distribution over the underlying true state.

**Contributions:**

- In a significant departure from previous research, we address the partial observability present in adversarial RL problems (due to adversarial perturbations) by introducing the concept of Adversarial Counterfactual Error (ACoE), which is defined based on beliefs about the underlying true state rather than the observable state only.

- We introduce a scalable surrogate for ACoE called Cumulative ACoE (C-ACoE) and establish its fundamental theoretical properties, which aid in developing strong solution methods.

- We develop mechanisms to minimize C-ACoE while maximizing expected value by leveraging established techniques from Deep Reinforcement Learning (e.g., DQN, PPO).

- Finally, we present comprehensive experimental results on benchmark problems (MuJoCo, Atari, Highway) employed in adversarial RL area to demonstrate the effectiveness of our approaches compared to leading methods (e.g., Protected, RADIAL, RAD, WOCAR) for adversarial reinforcement learning. We test against potent myopic attacks (such as MAD, PGD) and more advanced macro-strategic adversaries such as PA-AD (Sun et al., 2023).

## 2 RELATED WORK

**Adversarial attacks in RL**: Deep RL is vulnerable to attacks on the input, ranging from methods targeting the underlying DNNs such as an FGSM attack (Huang et al., 2017; Goodfellow et al., 2014), tailored attacks against the value function (Kos and Song, 2017; Sun et al., 2020), or adversarial behavior learned by an opposing policy (Gleave et al., 2019; Everett et al., 2020; Oikarinen et al., 2021; Zhang et al., 2020). We compile attacks on RL loosely into two groups of learned adversarial policies: observation poisonings (Gleave et al., 2019; Sun et al., 2020; Lin et al., 2017; Guo et al., 2023) and direct ego-state disruptions (Pinto et al., 2017; Rajeswaran et al., 2017). Each category has white-box counterparts that leverage the victim's network gradients to generate attacks (Goodfellow et al., 2014; Oikarinen et al., 2021; Huang et al., 2017; Everett et al., 2020). In this work (similar to existing works highlighted in this section), we focus on defending against the former group, observation poisonings, with both white-box and black-box scenarios.

**Adversarial Retraining and Adversary Agnostic Approaches**: In adversarial retraining, adversarial examples are found or generated and integrated into the set of training inputs (Shafahi et al., 2019; Ganin et al., 2016; Wong et al., 2020; Madry et al., 2017; Andriushchenko and Flammarion, 2020; Shafahi et al., 2020). For a comprehensive review, we refer readers to Bai et al. (2021). In RL, research efforts have demonstrated the viability of training RL agents against adversarial examples (Gleave et al., 2019; Bai et al., 2019; Pinto et al., 2017; Tan et al., 2020; Kamalaruban et al., 2020; Sun et al., 2023). Training RL agents against known adversaries is a sufficient defense against known attacks; there are effective adversarial retraining methods grounded in many disciplines such as curriculum learning (Wu and Vorobeychik, 2022), policy-adversary training (Sun et al., 2023) and behavior cloning (Nie et al., 2024). However, novel or more general adversaries remain effective against this class of defense (Gleave et al., 2019; Kang et al., 2019). Furthermore, they often take longer to train (needing to train both victim and adversary policies). The adversarial retraining technique PA-ATLA-PPO (Sun et al., 2023) reports needing 2 million training frames for MuJoCo-Halfcheetah. For comparison, both RAD (Belaire et al., 2024) and WocaR-PPO (Liang et al., 2022) are adversary-agnostic methods, and require less than 40% of the training frames. This paper focuses on adversary-agnostic defenses that do not train against specific adversaries in the environment.

**Robust Regularization**: Regularization approaches (Zhang et al., 2020; Oikarinen et al., 2021; Everett et al., 2020) take vanilla value-optimized policies and robustify them to minimize the loss due to adversarial perturbations. These approaches utilize certifiable robustness bounds computed for neural networks when evaluating adversarial loss and ensure the probability an attacker successfully changes the agent's actions is reduced using these lower bounds. Despite lowering the likelihood of a successful attack, a successful attack (i.e., two close states have different actions creates vulnerability) is still just as effective. Previous works suggest the need to learn safe trajectories via robustness-specific objectives, rather than a robust decision classifier only (Belaire et al., 2024; Liang et al., 2022; Li et al., 2024), such that successful attacks (if any) are also less effective.

**Robust Control**: Measuring and optimizing a regret value to improve robustness has been studied previously in uncertain Markov Decision Processes (MDPs)(Ahmed et al., 2013; Rigter et al., 2021; Adulyasak et al., 2015). In RL, Jin et al. (2018) establish Advantage-Like Regret Minimization (ARM) as a policy gradient solution for agents robust to partially observable environments. In continuous time control, Yang et al. (2023) study the composition of robust control algorithms with a robust predictor of perturbed system dynamics. In contrast to policy regret, we form beliefs about true states and minimize the cumulative adversarial counterfactual error (a novel notion of action-regret) to ensure a robust policy is computed, also recognizing the partial observability present in the problem.

**Game Theoretic Approaches**: A thread of approaches (McMahan et al., 2024; Liang et al., 2024) have employed partially observable stochastic games to represent problems of interest. A key advantage of game theoretic approaches is their ability to reason about adversaries. However, they assume that an adversary is always present–this can result in conservative solutions–and typically are computationally heavy. We do not use equilibrium concepts to ensure there is a good balance between robustness and "non-adversarial value" maximization. Instead, our risk-reward balance is computed based on the empirical belief about the adversary obtained from observations.

**Partially Observable Adversaries**: Several prior works (Jin et al., 2018; Zhang et al., 2020; Liu et al., 2024) have acknowledged and consider that adversarial observation perturbations make the underlying state partially observable. This has resulted in improved results. However, there are a few fundamental differences on how partial observability is considered in the most recent work (Liu et al., 2024) and our contributions:

- Partial observability is captured using a history of observations that does not consider that this partial observability is being driven by an adversary (i.e., with intention). The partial observability present in adversarial RL is not the same as in Partial Observable MDPs, where partial observability is a facet of the agent sensor (that is only stochastic, not adversarial). In our work, our belief state computation (to account for partial observability) explicitly considers that an adversary is driving the observation.

- In training, they compute a set of non-dominated policies to execute at test time. Then, they do test time adaptation, performing regret minimization over multiple (800) complete runs of the policy against the adversary. This is effective, though unfortunately impractical in domains such as autonomous vehicle control, where adapting to an adversary *after* a

catastrophe is not acceptable. Thus, such test time adaptation has not been utilized in any of the existing works, ours included.

- They do not adapt at every time step (which is feasible in RL settings based on observations), rather waiting until the end of each episode to adapt their policy meta-weights. Because time step-wise interaction and adaptation fits within RL settings, we consider the adversarial susceptibility of actions at every time step based on estimated belief and act accordingly.

## 3 ADVERSARIAL COUNTERFACTUAL ERROR (ACoE)

In this section, we define the ACoE objective for the Adversarial Reinforcement Learning (RL) problem. Intuitively, ACoE refers to the difference in the expected value obtained by a defender in the absence of adversarial perturbations versus in the presence of an adversary. It should be noted that in the case of adversarial perturbations, the defender only receives the altered state, and no information that is verified to be uncorrupted. By minimizing the ACoE objective in conjunction with maximizing expected value, we aim to derive a policy that provides a good trade-off between robustness (against adversary perturbations) and effectiveness (accumulating reward).

**Expected value without adversarial perturbations, $V(s)$:**

In the case without adversarial perturbations, the defender's problem is one of an infinite horizon MDP. Formally, we define the MDP $\langle \mathcal{S}, \mathcal{A}, T, R, \gamma \rangle$ where $\mathcal{S}$ is the state space, $\mathcal{A}$ is the action space, $T(s' \mid s, a)$ is transition probability, $R(s, a)$ is the immediate reward, and $\gamma$ is the discount factor. Without loss of generality, we assume $R(s, a) \in [0, 1]$. For ease of presentation, we assume discrete state and actions in the mathematical sections. The aim in the MDP is to choose actions at every time step (specified as a policy $\pi$) that maximize the value function $V$. In infinite horizon MDPs, the optimal policy is memoryless and stationary, i.e. a function of only the current state. However, to be more general and keep consistent notation with the case where there is an adversarial partially observable case below, we use $I$ as the current information state, i.e., $I$ is the sequence of observed states and actions up to the present, and the policy computes the action as a function of $I$, $\pi(I)$. Note that this is without loss of generality, as the optimal policy in an MDP will simply ignore the history preceding the current state. Then, the value for a policy $\pi$ is given by

$$V(s) = R(s, \pi(I)) + \gamma E_{s' \sim T(\cdot|s, \pi(I))}[V(s')]$$

**Expected value with adversarial perturbations, $U(b)$:**

In the case of an adversarial perturbation, the defender only receives an altered observation, providing only partial information about the underlying true state (i.e., the true state is near the perturbed state). Formally, we define the adversary's policy as a function, $\nu : \mathcal{S} \to \Delta(\mathcal{S})$, where $\Delta(\mathcal{S})$ denotes all possible distributions over $\mathcal{S}$; we also abuse notation slightly to indicate the perturbed random state as $\nu(s)$. We follow the standard assumption in adversarial learning that the perturbed state is close to the true underlying state, i.e., $||\nu(s) - s||_\infty \leq \epsilon$. This is an example of a one-sided Partially Observable Stochastic game (POSG) (Horák et al., 2023) in which the adversary has full observability while the defender does not observe the underlying state and only observes the perturbed state. It is well known (Horák et al., 2023) that with a fixed adversarial perturbation policy (possibly randomized), the defender's problem reduces to a Partially Observable Markov Decision Process (POMDP).

A POMDP is an MDP where the state is only partially observed. This partial observability is captured using an observation space $\mathcal{O}$ and observation probability $P_o(o \mid s', a)$ that specifies the probability of observing $o$ given true state $s'$ obtained on taking action $a$. Further, a POMDP is known to be equivalent to a belief state MDP (Kaelbling et al., 1998) where states are beliefs over the underlying states in the POMDP. A belief state, $b$ is a probability distribution over underlying states, s, where $\sum_s b(s) = 1$. On taking actions, this belief state changes and is computed by using a standard Bayesian update:

$$b'(s') = \frac{P_o(o \mid s', a) \sum_s T(s' \mid s, a) b(s)}{P_o(o \mid b, a)} \text{ where } P_o(o \mid b, a) = \sum_{s'} P_o(o \mid s', a) \sum_s T(s' \mid s, a) b(s)$$

We will employ a short form to represent the above update, $b' = SE(b, o, a)$. As the belief update requires knowledge of the model (transition function), our initial mathematical analysis is in a

model-based framework. An optimal policy in a POMDP can be a function of the belief. However, it is known that for POMDPs belief $b$ is a sufficient statistic for information state $I$, so we can consider the more general policy that depends on $I$, without any loss of generality. We denote by $U$ the value function of this POMDP for policy $\pi$:

$$U(b) = R(b, \pi(I)) + \gamma \sum_o P_o(o \mid b, \pi(I))U(SE(b, o, \pi(I)))$$

The partial observability exhibited in adversarial RL has a particular structure in which the observation space $\mathcal{O}$ is the same as the state space $\mathcal{S}$, and the observation probability function $P_o(o \mid s, a)$ is governed by the adversary's perturbation policy. More specifically, in our problem, the observation probability depends only on the true state and not the defender action, thus, we write $P_o^\nu(o \mid s)$, but note that $b' = SE(b, o, a)$ still depends on $a$ due to the use of transition $T$. Note that the non-adversarial case can be considered a special case where the adversary policy is the identity function id, and then $P_o^{\mathrm{id}}(o \mid s) = \mathbb{I}(o = s)$ for indicator function $\mathbb{I}$. As the observation space $\mathcal{O} = \mathcal{S}$, we will often use the notation $s_o$ to refer to an observation as $s_o \in \mathcal{S}$ where the subscript $o$ is used to denote that this is an observation. In particular, any distribution over the observation space is a distribution over the state space.

**Adversarial Counterfactual Error, ACoE**: We analyze the difference in return $V - U$ obtained in the non-adversary case (denoted by $V$) and adversary case (denoted by $U$) using a common policy $\pi$ in each case. We term $V - U$ as Adversarial Counterfactual Error (ACoE). As the optimal policy depends on different information structures in these two cases, to compare these cases with the same policy we have already chosen to generalize the policy as a function of the information state $I$. We write the value functions starting with the currently observed belief, where the non-adversarial case is the true state itself. For notational ease in the later sections, we will write $s_o$ to represent the current observation, which particularly emphasizes that in our problem, the observations are themselves part of the state space. Further, in our particular domain, $o \in \mathcal{S}$, thus, $P_o(\cdot \mid b, \pi(I))$ specifies a probability distribution over states. Thus, by renaming variables and dropping the dependence of observations on actions, we rewrite $\sum_o P_o(o \mid b, \pi(I))U(SE(b, o, \pi(I)))$ as $E_{s_o' \sim P_o(\cdot \mid b, \pi(I))}[U(SE(b, s_o', \pi(I))]$. Then, for both the non-adversary and adversary scenarios, following standard MDP and POMDP facts, we have a recursive form as below:

$$V(s_o) = R(s_o, \pi(I)) + \gamma E_{s_o' \sim T(\cdot \mid s_o, \pi(I))}[V(s_o')]$$
$$U(b) = R(b, \pi(I)) + \gamma E_{s_o' \sim P_o(\cdot \mid b, , \pi(I))}[U(SE(b, s_o', \pi(I))]$$

ACoE is defined as $V(s_o) - U(b)$.

We also use an additional shorthand notation of $T_o(\cdot, \cdot \mid b, a)$ to denote the joint probability distribution of $s_o'$ and $b'$ specified by the sampling process: $s_o' \sim P_o(\cdot \mid b, a), b' = SE(b, s_o', a)$. We define the following important quantity:

**Definition 3.1** (Cumulative Adversarial Counterfactual Error (C-ACoE)). Define C-ACoE as

$$\delta(s_o, b) = R(s_o, \pi(I)) - R(b, \pi(I)) + \gamma E_{s_o', b' \sim T_o(\cdot, \cdot \mid b, \pi(I))}[\delta(s_o', b')] \tag{1}$$

**Theorem 3.2.** *Let $K = \max_{s \in \mathcal{S}} V(s)$ and assume $TV(T(\cdot \mid s_o, a), P_o(\cdot \mid b, a)) \leq \Xi$ for any observed state $s_o$, belief $b$, and action $a$ in the same time step, then*

$$\left| V(s_o) - U(b) - \delta(s_o, b) \right| \leq \frac{\gamma K \Xi}{1 - \gamma}$$

The above result shows that there are two parts to ACoE, the uncontrollable part with the $TV$ distance captures structural differences in the transition without attack and transition induced by the attack, while the controllable part, C-ACoE term $\delta(s_o, b)$ captures long term return difference due to the adversarially induced transition. In the appendix, we delve more into the structural difference in transitions by utilizing Wasserstein distance instead of Total Variation, TV distance. The above results also suggest that apart from the inherent structural differences, minimizing C-ACoE $\delta(s_o, b)$ can be effective in ensuring that returns in the adversarial scenario are close to the non-adversarial scenario, which we explore in the next section.

Since the structural differences in transition are not controllable by the defender agent, we focus on minimizing the C-ACoE for the defender. Furthermore, to ensure that the effectiveness of the policy in accumulating rewards is high, we minimize C-ACoE while simultaneously maximizing the non-adversarial expected reward.

---

**Algorithm 1:** $\delta$-PPO

---

**1** Initialize policy network weights $\theta_1$, value network weights $\phi_1$, and $\delta$-network weights $\psi_1$

**2** Set robustness-hyperparameter $\lambda$

**3** **for** *iteration* $k \in \{1, \dots, M\}$ **do**

**4**      Collect set of trajectories $\mathcal{D}_k$ by running policy $\pi_{\theta_k}$ multiple times for $T$ steps

**5**      Estimate rewards-to-go $\hat{R}_t$ and C-ACoE-to-go $\hat{\delta}_t$ at all time steps $t$ for all trajectory in $\mathcal{D}_k$

**6**      Compute advantage estimates $\hat{A}$ using Generalized Advantage Estimator (Schulman et al., 2016), based on $\hat{R}_t$'s and $V_{\phi_k}$

**7**      Compute C-ACoE Advantage $A_{c,t} = \hat{A}_t - \lambda\hat{\delta}_t$

**8**      Update policy parameters to $\theta_{k+1}$ by maximizing the PPO-clipped (Schulman et al., 2017) form of $A_{c,t}$

**9**      Update $\phi_{k+1} = \text{argmin}_\phi \frac{1}{|\mathcal{D}_k|T} \sum_{\tau \in D_k} \sum_{t=0}^T (V_\phi(s_t) - \hat{R}_t)^2$

**10**      Update $\psi_{k+1} = \text{argmin}_\psi \frac{1}{|\mathcal{D}_k|T} \sum_{\tau \in D_k} \sum_{t=0}^T (\delta_\psi(s_t) - \hat{\delta}_t)^2$

---

## 4   OPTIMIZING C-ACoE ALONG WITH NON-ADVERSARIAL EXPECTED REWARD IN ADVERSARIAL RL

In RL settings, we do not have the model and hence the transition dynamics $T$ are not available. Thus, computing $\delta(s_o, b)$ exactly is not possible, as the belief depends on knowledge of transition probabilities. However, our problem presents a structured scenario where the observation depends only on the current true state and uncertainty is entirely due to adversarial perturbation. It has been stated in literature and is also intuitive that adversarial perturbations are effective in causing harm when they induce a large enough change in the defender's action distribution (Oikarinen et al., 2021; Zhang et al., 2020). Thus, we propose to derive a surrogate belief based on the observed state $s_o$ in conjunction with reasoning about how the adversary might have forced this observation to arise. We present a couple of such belief constructions here.

Using the full history of observations and actions (represented as the information state, $I$) as an input to the policy is computationally expensive to implement. Prior approaches have used a variety of approximations (Azizzadenesheli et al., 2018); we adopt a simple measure (Müller and Montufar, 2021; Kober et al., 2013) where we restrict solutions to the set of policies that depend just on the current observation. Next, note that if $b$ depends on $s_o$ only, then $\delta(s_0, b)$ is a function of $s_o$ only. Hence, we redefine the C-ACoE as

$$\delta(s_o) = R(s_o, \pi(s_o)) - R(b(s_o), \pi(s_o)) + \gamma E_{s'_o \sim \nu(s'), s' \sim T(\cdot \mid s, \pi(s_o))}[\delta(s'_o)] \quad (2)$$

We note that the underlying true state $s'$ is not observed, but estimating the second term on the RHS above requires only samples of observation $s'_o$ which are available from the simulator. In this form, C-ACoE also satisfies the Bellman optimality structure (as stated formally in the following proposition) and hence allows for incorporating the minimization of $\delta(s_o)$ in standard RL techniques.

**Proposition 4.1.** *Let $\delta^*(s_o)$ be the minimum C-ACoE value from observation $s_o$. Then,*

$$\delta^*(s_o) = \min_a \{R(s_o, \pi(s_o)) - R(b(s_o), a) + \gamma E_{s'_o \sim \nu(s'), s' \sim T(\cdot \mid s, a)}[\delta^*(s'_o)]\}$$

Algorithm 1 shows our adaptation of PPO for optimizing $\delta$ along with maximizing $V$. The steps for maximizing $V$ follow standard steps in PPO leading to the standard advantage $\hat{A}_t$ in line 7. We also compute the C-ACoE-to-go from the sampled trajectories (line 5) and use it to augment the standard advantage $\hat{A}_t$ in line 7 (we need to minimize C-ACoE, hence the negative sign before $\hat{\delta}_t$). Line 9 is a standard PPO step to update the $V$ network and we do so similarly for the $\delta$ network in line 10. We found that computing an advantage-like term for $\delta$ did not improve performance, thus we used only C-ACoE-to-go. A similar adaptation is also done for DQN, presented in the appendix. Next, we describe two possible belief constructions given the observed state $s_o$.

**Adversary-Aware Belief Estimation (A2B)**: We aim to assign a belief to states in neighborhood $N(s_o)$ of observation, $s_o$ where $N(s_o) = \{s \mid ||s - s_o|| \leq \epsilon\}$. $N(s_o)$ is restricted to an $\epsilon$ bound

given established adversarial perturbation practices. We know that an adversarial perturbation from state $s$ to state $s_o$ is an effective attack when the action distribution $\pi(s)$ and $\pi(s_o)$ are quite different. Based on this fact, we form a belief:

$$b(s) = \frac{e^{D_{KL}(\pi(s)||\pi(s_o))}}{\sum_{s' \in N(s_o)} e^{D_{KL}(\pi(s')||\pi(s_o))}}$$

**Adversary-Attack-Aware Belief Estimation (A3B)**: Different from A2B, we assign scores to states in $N(s_o)$ based on assumptions about adversarial preference. These scores depend on a surrogate attack $\nu$, for which we use a 50-step PGD attack; quick empirical checks show this to find the worst-case bound of the $L_\infty$-norm ball in nearly every state. We assign a score $z(s)$ to a state $s \in N(s_o)$ that is a ratio of: (the KL divergence of the action distributions at possibly perturbed observation $s_o$ and the state $s$) to (the KL divergence of actions distribution at $\nu(s)$ and $s$). Then, a belief is assigned to state $s'$ depending on the score $z$ by a softmax operation:

$$b(s) = \frac{e^{z(s)}}{\sum_{s' \in N(s_o)} e^{z(s')}} \quad \text{where} \quad z(s) = \frac{D_{KL}(\pi(s_o)||\pi(s))}{D_{KL}(\pi(\nu(s))||\pi(s))}$$

The intuition for the above formulation of score $z$ is that if the true state was $s$, the adversary should prefer to provide $\nu(s)$ with a high KL divergence between action distributions at $\nu(s)$ and $s$, but since we observed $s_o$, the ratio of KL divergences in score $z(s)$ measures how effective the change $s$ to $s_o$ is, compared to the change $s$ to $\nu(s)$. Any candidate true state $s$ has low score if $s_o$ is *not* an effective attack from state $s$. Thus, A3B reduces the scores (weights) of states that are unlikely adversarial choices based on the policy $\pi$. Then, optimizing C-ACoE using A3B beliefs coupled with non-adversarial value maximization allows balancing unperturbed performance with robustness, as highlighted earlier in the introduction.

For a visual explanation of the logic of A3B, consider Figure 1. This figure shows two neighborhood states $s_1$ and $s_2$ which could potentially be the underlying true state, given the observed state $s_o$. Subsequently, $N(s_1)$ contains a worst-PGD perturbation $s_1' = \nu(s_1)$ and $N(s_2)$ similarly contains $s_2' = \nu(s_1)$. Even though $s_2'$ may be close in Euclidean distance to $s_o$, it is possible that

$$D_{KL}(\pi(s_2')||\pi(s_2)) \gg D_{KL}(\pi(s_o)||\pi(s_2))$$

leading to a small score $z_{s_2}$ (closer to 0) for $s_2$. This is intuitive, as an adversary will likely not perturb $s_2$ to $s_o$, due to the existence of the more disruptive attack $s_2'$. Similarly, the score $z_{s_1}$ for $s_1$ can be close to 1 due to $D_{KL}(\pi(s_1')||\pi(s_1)) \approx D_{KL}(\pi(s_o)||\pi(s_1))$, which is intuitive as $s_o$ results in same amount of change in action distribution as $s_1'$.

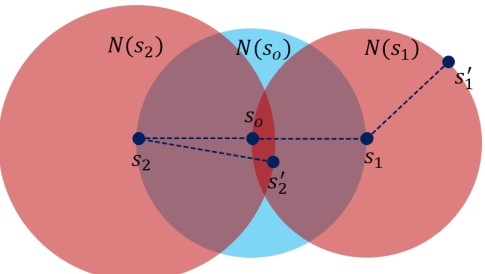

Figure 1: A3B belief construction. Let the dotted line $\overline{s_i s_j}$ have magnitude representing the damage when perturbing $s_i \to s_j$. In this example, our method should discount the possibility that $\nu(s_2) = s_0$, and lessen the score $z(s_2)$.

**Continuous State Sampling**: One issue to consider above is when the state space is continuous. In such a scenario, we still form a finite set $N(s_o)$ by uniformly sampling a given number $n$ (hyperparameter) of samples from the continuous set $C = \{s \mid ||s - s_o|| \leq \epsilon\}$. From the definition of $\delta$ (Eq. 2), we use $b$ to estimate $R(b, a)$. Our true value of this is $R = R(b, a) = \int_{s \in C} R(s, a) p(s) ds$ where the probability density $p(s) = e^{z(s)} / \int_{s \in C} e^{z(s)} ds$. In contrast, we sample $n$ states from a uniform distribution $U$ with probability density given by $u(s) = 1/vol(C)$ where $vol$ is the volume of set $C$ and estimate $\hat{R} = \frac{\sum_{s' \in N(s_o)} R(s', a) e^{z(s')}}{\sum_{s' \in N(s_o)} e^{z(s')}}$. We show a result in the appendix that justifies the estimate $\hat{R}$ by showing that the expected value of this estimate is close to the true required value $R$.

**Recurrent State History**: A3B is primarily proposed as an adversary-aware method of deriving belief about true states based on the current observation. However, this approach can be adapted to consider a history of observations, albeit with higher computational burden. We provide an extended A3B definition with multistep observations and additional evaluations of this extended A3B using an LSTM network in the Appendix.

Table 1: Experimental results versus myopic adversaries. Each row shows the mean scores of each RL method against different attacks. The most robust scores are in **bold**. Our approaches are A2B and A3B, which are highlighted .

| Method | Unperturbed | MAD | PGD | Unperturbed | MAD | PGD |
|---|---|---|---|---|---|---|
| | highway-fast-v0 | | | merge-v0 | | |
| PPO | 24.8±5.42 | 13.63±19.85 | 15.21±16.1 | 14.94±0.01 | 10.2±0.02 | 10.42±0.95 |
| CARRL | 24.4±1.10 | 4.86±15.4 | 12.43±3.4 | 12.6±0.01 | 12.6±0.01 | 12.02±0.01 |
| RADIAL | 28.55±0.01 | 2.42±1.3 | 14.97±3.1 | 14.86±0.01 | 11.29±0.01 | 11.04±0.91 |
| WocaR | 21.49±0.01 | 6.15±0.3 | 6.19±0.4 | 14.91±0.04 | 12.01±0.28 | 11.71±0.21 |
| RAD | 21.01±0.01 | 20.59±4.1 | 20.02±0.01 | 13.91±0.01 | 13.90±0.01 | 11.72±0.01 |
| A2B | 24.8±0.01 | 23.11±0.01 | 20.8±12.6 | 14.91±0.01 | 14.23±0.8 | 12.92±0.13 |
| A3B | 23.8±0.01 | **23.21±0.01** | **22.61±14.1** | 14.91±0.17 | **14.88±0.17** | **14.89±0.17** |

# 5 EXPERIMENTS

We provide empirical evidence to show the effectiveness of our proposed method. In particular, we want to investigate whether A2B and A3B improve over leading adversarial robustness methods on established baselines, and what aspects of C-ACoE contribute to a viable defense against strategic adversaries.

## 5.1 EXPERIMENT SETUP

We evaluate C-ACoE methods on the standard Atari (Bellemare et al., 2013) and MuJoCo (Todorov et al., 2012) domains, and additionally the Highway simulators (Leurent, 2018), to demonstrate real problems of interest. In the Mujoco and Highway tasks, the agent earns score by traversing distance without incurring critical collisions. Atari tasks are game-dependent. We use a standard training setup seen in (Oikarinen et al., 2021; Liang et al., 2022; Belaire et al., 2024), and detailed in Appendix C.

We compare C-ACoE optimization methods (A2B, A3B) to the following baselines: PPO (Schulman et al., 2017); CARRL, a simple but robust minimax method (Everett et al., 2020); RADIAL, a leading regularization approach (Oikarinen et al., 2021); WocaR, worst-case aware value maximization (Liang et al., 2022); RAD, a method minimizing a notion of regret (Belaire et al., 2024); and Protected (Liu et al., 2024). We test all methods against two greedy attack approaches of reward-minimizing policy adversaries and gradient attacks. We evaluate each method's PPO implementation in the Highway and Mujoco domains, and DQN implementations in Atari tasks. Additional comparisons including to a few more baselines, namely BCL (Wu and Vorobeychik, 2022) and CAR-DQN (Li et al., 2024), are in the Appendix.

**Protected Baseline**: We wish to specifically address the comparison with Protected (Liu et al., 2024). Protect does regret minimization (EXP3) over multiple rounds (each round is full policy episode) and the weights are updated *at test time* based on empirical return in each round. As stated earlier, this has a major advantage against all other approaches in the literature, which do not do any test time adaptation, and unfortunately makes Protected impractical for safe RL applications. To indicate this, the results of the original Protected are presented but grayed out (and not compared to when highlighting best result) in Table 3. The test time adaptation also results in Protected having a significantly higher unperturbed score in some of the domains (e.g., HalfCheetah, Walker2d, Ant) even when compared to PPO. Therefore, for a *fair comparison* to all the adversarial RL approaches, we also provide a comparison against a variant of Protected, referred to as Protected[†], where there is no test time adaptation. Further details of Protected and additional comparison is presented in Appendix.

**Myopic Adversaries**: We test the adversarial robustness of each method against adversaries that we term as "greedy" or myopic, meaning that they compute worst-case attacks for a given time step. Following the set up employed in existing works, we measure a 10-step PGD attack (Madry et al., 2017) with $\epsilon = 0.1$, and a MAD attack (Zhang et al., 2020) with $\epsilon = 0.15$. We evaluate both MAD and PGD attacks as they represent two distinct attack directions (MAD is reward-based, while PGD is a gradient-based).

Table 2: Experimental results versus myopic adversaries in Atari domains, formatted the same as Table 1. Methods are evaluated as their corresponding DQN implementations.

| Method | Unperturbed | MAD | PGD | Unperturbed | MAD | PGD |
|---|---|---|---|---|---|---|
| | Pong | | | Freeway | | |
| PPO | 21.0±0 | -20.0±0.07 | -19.0±1.0 | 29 ± 3.0 | 4 ± 2.31 | 2±2.0 |
| CARRL | 13.0 ±1.2 | 11.0±0.010 | 6.0±1.2 | 18.5±0.0 | 19.1 ±1.20 | 15.4±0.22 |
| RADIAL | 21.0±0 | 11.0±2.9 | **21.0± 0.01** | 33.2±0.19 | 29.0±1.1 | 24.0±0.10 |
| WocaR | 21.0±0 | 18.7 ±0.10 | 20.0 ± 0.21 | 31.2±0.41 | 19.8±3.81 | 28.1±3.24 |
| RAD | 21.0±0 | 14.0 ± 0.04 | 14.0 ± 2.40 | 33.2±0.18 | 30.0±0.23 | 27.7±1.51 |
| A2B | 21.0±0 | 20.1±0.04 | **21.0±0.01** | 33.2±0.18 | 30.1±0.43 | **30.8±1.51** |
| A3B | 21.0±0 | **20.8±0.7** | **21.0±0.01** | 33.2±0.18 | **31.0±0.87** | **31.1±1** |

Table 3: Experimental results versus myopic adversaries in Mujoco domains, formatted the same as Table 1. Methods are evaluated as their corresponding PPO implementations. Note: the Protected method requires test time adaptation rounds to achieve full results. The Protected method without test time adaptation is labelled as *Protected*[†].

| Method | Unperturbed | MAD | PGD | Unperturbed | MAD | PGD |
|---|---|---|---|---|---|---|
| | Hopper | | | Walker2d | | |
| PPO | 4128 ± 56 | 1110±32 | 128±105 | 5002 ± 20 | 680±1570 | 730±262 |
| RADIAL | 3737±75 | 2401±13 | 3070±31 | 5251±10 | 3895±128 | 3480±3.1 |
| WocaR | 3136±463 | 1510 ± 519 | 2647 ±310 | 4594±974 | 3928±1305 | 3944±508 |
| Protected | 3652±108 | 2512±392 | 2221± 775 | 6319±31 | 5148±1416 | 4720± 1508 |
| Protected[†] | 3573±81 | 2398±665 | 2215±98 | 5019 ± 87 | 3887 ± 492 | 3613 ± 487 |
| RAD | 3473±23 | 2783±325 | 3110±30 | 4743±78 | 3922±426 | 4136±639 |
| A2B | 3710±11 | 3240±41 | 3299±28 | 4760±61 | 4636±87 | 4708±184 |
| A3B | 3766±23 | **3370±275** | **3465±17** | 5341±60 | **5025±94** | **5292±231** |
| | HalfCheetah | | | Ant | | |
| PPO | 5794 ± 12 | 1491±20 | -27±1288 | 5620±29 | 1288±491 | 1844±330 |
| RADIAL | 4724±76 | 4008±450 | 3911±129 | 5841±34 | 3210±380 | 3821±121 |
| WocaR | 5220±112 | 3530±458 | 3475±610 | 5421±92 | 3520±155 | 4004±98 |
| Protected | 7095±88 | 4792±1480 | 4680±1203 | 5769±290 | 4440±1053 | 4228± 484 |
| Protected[†] | 4777±360 | 4551±843 | 3997±285 | 4620±32 | **4264±166** | 4368±473 |
| RAD | 4426±54 | 4240±4 | 4022±851 | 4780±10 | 3647±32 | 3921±74 |
| A2B | 5192 ±56 | 4855± 120 | 4722±33 | 5511±13 | 3824±218 | 4102±315 |
| A3B | 5538±20 | **4986±41** | **5110±22** | 5580±41 | 4071±242 | **4418±290** |

**Long-Horizon Adversaries**: We also assess adversarial robustness of each method versus more strategic, long-horizon adversaries that compute worst-case *trajectories* to deceive an RL agent. We evaluate agents against PA-AD (Sun et al., 2023), the state-of-the-art adversarially-directed policy attack, as well as the Critical Point Attack (Liang et al., 2022) and Strategically Timed Attack (Lin et al., 2017). We evaluate the adversarial robustness of the target policies as the depth of strategy increases for the long-horizon adversaries. In the context of the Critical Point attack, a higher depth of strategy increases the length and number of trajectories sampled to find the worst-case future outcome, and a stronger Strategically Timed attacker has a larger perturbation budget.

## 5.2 RESULTS

In Tables 1, 2, and 3, we report the mean result over 5 policies initialized with random seeds, with 50 test episodes each. The variance reported ($\pm\sigma$) is the standard deviation from the mean for each method. The most robust score is shown in **boldface**.

**Myopic attacks**: As seen in Table 1-3, C-ACoE methods A2B and A3B achieve state of the art robust performance against standard greedy attacker strategies, as well as nominal performance similar to the

Table 4: Robust performance against the PA-AD attacker (Sun et al., 2023). We train the attacker with the PA-AD framework against the completed victim policies for 500 episodes, the same for each victim and environment. As the Protected method has several PA-AD attackers (for each non-dominated policy), we instead use the sampling schema outlined in their work.

| Method | PA-AD Perturbed Scores | | | |
| | HalfCheetah | Walker2d | Hopper | Ant |
|---|---|---|---|---|
| PPO | $-388 \pm 820$ | $427 \pm 32$ | $167 \pm 93$ | $-121 \pm 1255$ |
| Radial | $3441 \pm 42$ | $3703 \pm 202$ | $2288 \pm 74$ | $2567 \pm 41$ |
| Wocar | $4148 \pm 68$ | $3895 \pm 126$ | $2387 \pm 114$ | $2779 \pm 170$ |
| Protected | $4411 \pm 718$ | $5803 \pm 857$ | $2896 \pm 723$ | $4312 \pm 281$ |
| Protected$^\dagger$ | $2331 \pm 277$ | $4480 \pm 492$ | $2210 \pm 385$ | $3103 \pm 96$ |
| RAD | $4233 \pm 13$ | $3864 \pm 67$ | $2403 \pm 129$ | $2756 \pm 81$ |
| A2B | $4393 \pm 79$ | $3997 \pm 214$ | $2441 \pm 31$ | $2821 \pm 312$ |
| A3B | $\mathbf{4478 \pm 67}$ | $\mathbf{4931 \pm 166}$ | $\mathbf{2580 \pm 92}$ | $\mathbf{3205 \pm 275}$ |

best observed value-maximizing methods such as PPO. We attribute this success to the two parts of ACoE: framing the adversarial robustness problem as a POMDP and the simultaneous maximization of value and *minimization of ACoE error* brings increased performance over maximin methods and higher robustness overall. Our approaches perform better than Protected with test time adaptation and also and Protected$^\dagger$ in all the cases, except Ant.

**Long-horizon attacks**: We also test our methods against attackers with a longer planning horizon (and not only the myopic attackers from above). In Figure 4 and Table 4, we test the performance of our approaches in the presence of the SOTA attack, referred to as the PA-AD policy attack (Sun et al., 2023). We also include experiments evaluating robust methods against the Strategically Timed attack (Lin et al., 2017) and the Critical Point attack(Sun et al., 2020) in the appendix. We find that across domains, C-ACoE agents maintain robustness even against long-horizon attacks. This is one of the main advantages of our proposed methods following the C-ACoE-minimizing philosophy, as the error-robust policies seek stable trajectories rather than robust single-step action distributions.

**Robust Behavior**: In Appendix Figure 5, we observe qualitative differences between PPO, A3B, and WocaR. The WocaR agent adopts more stable motion, minimizing the worst-case, and PPO optimizes for speed, only using the back leg. A3B balances the two approaches, using both legs to keep stability while still retaining a wide range of motion. Full videos of the behaviors described in Figure 5 can be viewed from DropBox at tinyurl.com/a3b-gif, where the extent of robust behavior can be better observed.

# 6 DISCUSSION AND LIMITATIONS

We introduce the novel concept of ACoE based on beliefs about true state. We propose a scalable approximation of ACoE, C-ACoE, and demonstrate its usefulness in proactive adversarial defense, achieving state of the art robustness against strong observation attacks from both greedy and strategic adversaries on a variety of benchmarks. More importantly, we find that recognizing the partially observable nature of the defender agent in adversarial RL problems and optimizing ACoE can be used to increase the robustness of RL to adversarial observations, even against stronger or previously unseen attackers. In this paper, we focused on the estimation of belief states from single step perturbed observations; It may be beneficial to further estimate belief based on observations over multiple time steps. Some preliminary results on this are in the appendix, and addressing the computational complexity of multistep observation based belief construction makes for promising future work. We also note that the efficacy of the belief construct that we use is reliant on the accuracy of using KL Divergence as a notion of attack strength. We find our measures to be empirically the strongest, compared to notions such as Euclidean state distance, other F-divergences, or minimum reward, however, and leave other more complex measures to future work.

ETHICS STATEMENT

By trying to understand how to produce robust and safe RL policies, we unavoidably create knowledge on the destruction of prior policies. While this pursuit yields a net positive result by far, it is still important to acknowledge the risks associated with this field of research. In this paper specifically, we acknowledge the information asymmetry between the attacker and defender in the problem, as well as the insight that an adversary is, in general, considering attacks that change the victim's behavior to the greatest extent. These insights are formal definitions of existing dynamics, and while their acknowledgement may yield some tools to bad actors, we also provide formal and explicit tools to mitigate those harms.

REPRODUCIBILITY

We have uploaded code as part of our submission, showcasing the implementation of our ACoE-optimizing PPO methods, as well as the computation of A3B and A2B. Additionally, Algorithm 1 and 2 provide pseudocode-level instructions on the implementation of our methods. We have listed hyperparameter values and additional details in the appendix. All proofs in our paper are also present in the appendix.

ACKNOWLEDGMENTS

This research/project is supported by the National Research Foundation Singapore and DSO National Laboratories under the AI Singapore Programme (AISG Award No: AISG2-RP-2020-017) and the grant W911NF-24-1-0038 from the US Army Research Office.

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

## A PROOFS AND ADDITIONAL THEORY RESULTS

*Proof of Theorem 3.2.* Subtracting $U$ from $V$, and adding and subtracting $\gamma E_{s'_o \sim P_o(\cdot \mid b, \pi(I))}[V(s'_o)]$ we get

$$V(s_o) - U(b) =$$
$$R(s_o, \pi(I)) - R(b, \pi(I)) + \gamma E_{s'_o \sim P_o(\cdot \mid b, \pi(I))}[V(s'_o) - U(b')]+$$
$$\gamma E_{s'_o \sim T(\cdot \mid s_o, \pi(I))}[V(s'_o)] - \gamma E_{s'_o \sim P_o(\cdot \mid b, \pi(I))}[V(s'_o)]$$

Note that by definition of $T_o$, we have that $E_{s'_o \sim P_o(\cdot \mid b, \pi(I))}[V(s'_o) - U(b')] = E_{s'_o, b' \sim T_o(\cdot, \cdot \mid b, \pi(I))}[V(s'_o) - U(b')]$

Next, from Holder's inequality, we get that

$$\left| E_{s'_o \sim T(\cdot \mid s_o, \pi(I))}[V(s'_o)] - E_{s'_o \sim P_o(\cdot \mid b, \pi)}[V(s'_o)] \right| \leq \max_s \{V(s)\} TV(T(\cdot \mid s_o, \pi(I), P_o(\cdot \mid b, \pi(I)))) \tag{3}$$

Thus, for one side of the inequality above (i.e., using $a \leq b$ from the shown $|a| \leq b$, the other side is $-b \leq a$)

$$V(s_o) - U(b) \leq$$
$$R(s_o, \pi(I)) - R(b, \pi(I)) + \gamma E_{s'_o, b' \sim T_o(\cdot, \cdot \mid b, \pi(I))}[V(s'_o) - U(b') + \gamma K\Xi$$

For notation simplicity, let $R(s, \pi(I)) - R(b, \pi(I)) = \delta_R(s, b)$. We use $I'$ as the updated information state obtained by concatenating $I$ with $\pi(I), s'_o$. Applying the above recursively, we get

$$V(s_o) - U(b)$$
$$\leq \delta_R(s_o, b) + +\gamma E_{s'_o, b' \sim T_o(\cdot, \cdot \mid b, \pi(I))}[V(s'_o) - U(b')] + \gamma K\Xi$$
$$\leq \delta_R(s_o, b) + \gamma E_{s'_o, b' \sim T_o(\cdot, \cdot \mid b, \pi(I))}\left[\delta_R(s'_o, b') + \gamma E_{s''_o, b'' \sim T_o(\cdot, \cdot \mid b', \pi(I'))}[V(s''_o) - U(b'')] + \gamma K\Xi\right] + \gamma K\Xi$$
$$\leq \dots$$
$$\leq E_{(s_o, b, s'_o, b', \dots) \sim \pi, T, P_o}[\delta_R(s_o, b) + \gamma \delta_R(s'_o, b') + \gamma^2 \delta_R(s''_o, b'') + \dots] + \frac{\gamma K\Xi}{1 - \gamma}$$

We note that $E_{(s_o, b, s'_o, b', \dots) \sim \pi, T, P_o}[\delta_R(s_0, b) + \gamma \delta_R(s'_o, b') + \gamma^2 \delta_R(s''_o, b'') + \dots] = \delta(s_o, b)$, where

$$\delta(s_o, b) = R(s_o, \pi(I)) - R(b, \pi(I)) + \gamma E_{s'_o, b' \sim T_o(\cdot, \cdot \mid b, \pi(I))}[\delta(s'_o, b')]$$

Thus,

$$V(s_o) - U(b) \leq \delta(s_o, b) + \frac{\gamma K\Xi}{1 - \gamma}$$

By symmetric argument using other side of Eq. 3, we get

$$\delta(s_o, b) - \frac{\gamma K\Xi}{1 - \gamma} \leq V(s_o) - U(b)$$

These last two equations led to the statement in the theorem.

$\square$

The result above uses total variation distance (other work in literature also do (Zhang et al., 2020)), but, total variation is not as informative a distance measure as Wasserstein distance. For example, it is easy to see that $TV(P, Q) = 1$ whenever the support of $P$ and $Q$ do not overlap, but it does not distinguish whether the non-overlapping supports are near or far apart. As shown in prior work on WGAN (Arjovsky et al., 2017), Wasserstein distance provides more fine-grained distinctions. Also, the assumed bound $\Xi$ above hides the effect of the nature of the underlying transition $T$ on the bound. Hence, we prove the next result using Wasserstein distance, which reveals these facets of the problem.

**Theorem A.1.** *Assume that (1) $V$ is $L$-Lipschitz and (2) for any $\|s - s'\|_\infty \leq \epsilon$ and any action $a$ we have $W_1(T(\cdot \mid s, a), T(\cdot \mid s', a))) \leq \xi$. Then,*

$$\left| V(s_o) - U(b) - \delta(s_o, b) \right| \leq \frac{\gamma L(\xi + \epsilon)}{1 - \gamma}$$

*Proof of Theorem A.1.* The overall proof follows the same structure as Theorem 3.2. The only difference is in the bound for

$$\left| E_{s'_o \sim T(\cdot|s_o, \pi(I))}[V(s'_o)] - E_{s'_o \sim P_o(\cdot \mid b, \pi(I))}[V(s'_o)] \right| \tag{4}$$

For a simpler presentation, we use $a, a'$ to denote action taken in current and next time step. As $V/L$ is 1-Lipschitz, by duality of $W_1$ Wasserstein distance, we have

$$\left| E_{s'_o \sim T(\cdot|s_o, a)}[V(s'_o)/L] - E_{s'_o \sim P_o(\cdot \mid b, \pi(I))}[V(s'_o)/L] \right| \leq W_1(T(\cdot|s_o, a), P_o(\cdot \mid b, a))$$

or multiplying by $L$

$$\left| E_{s'_o \sim T(\cdot|s_o, a)}[V(s'_o)] - E_{s'_o \sim P_o(\cdot \mid b, \pi(I))}[V(s'_o)] \right| \leq L W_1(T(\cdot|s_o, a), P_o(\cdot \mid b, a))$$

Next, we bound $W_1(T(\cdot|s_o, a)$. Note that $P_o(\cdot \mid b, a)) = \sum_{s'} P_o^\nu(\cdot \mid s') \sum_s T(s' \mid s, a) b(s)$. First, because the restriction on adversarial perturbation, we know that if $b(s) > 0$ then $||s - s_o||_\infty \leq \epsilon$. Then, based on our assumption

$$W_1(T(\cdot|s_o, a), T(\cdot|s, a)) \leq \xi \text{ for any } s \text{ such that } b(s) > 0 \tag{5}$$

First, note that $W_1$ is a convex function of its argument. This can be seen easily; we show it for the first argument below. Recall that definition of $W_1(\mu, \nu) = \inf_{\gamma \in \Gamma(\mu, \nu)} \int d(x, y) \gamma(dx, dy)$ for couplings (joint distribution) set $\Gamma$ that have marginal as $\mu, \nu$. Choose $\gamma_1^*$ as a minimizer in $W_1(\mu_1, \nu)$ and $\gamma_2^*$ as a minimizer in $W_1(\mu_2, \nu)$. Let $\gamma^* = \alpha \gamma_1^* + (1 - \alpha) \gamma_2^*$; it easy to see that $\gamma^* \in \Gamma(\mu.\nu)$. Then,

$$\begin{aligned} W_1(\alpha \mu_1 + (1 - \alpha) \mu_2, \nu) &= \inf_{\gamma \in \Gamma(\alpha \mu_1 + (1 - \alpha) \mu_2, \nu)} \int d(x, y) d\gamma(x, y) \\ &\leq \int d(x, y) d\gamma^*(x, y) \\ &= \alpha \int d(x, y) d\gamma_1^*(x, y) + (1 - \alpha) \int d(x, y) d\gamma_2^*(x, y) \\ &= \alpha W_1(\mu_1, \nu) + (1 - \alpha) W_1(\mu_2, \nu) \end{aligned}$$

Let $T(\cdot \mid b, a) = \sum_s T(\cdot \mid s, a) b(s)$. Using the above convexity of $W_1$, we get that

$$W_1(T(\cdot|s_o, a), T(\cdot|b, a)) \leq \sum_s b(s) W_1(T(\cdot|s_o, a), T(\cdot|s, a)) \leq \xi \tag{6}$$

where the last inequality follows from Eq. 5

Next, we bound $W_1(P_o(\cdot \mid b, a), T(\cdot \mid b, a))$. First, by definition of $T(\cdot \mid b, a)$ we get that $P_o(\cdot \mid b, a) = \sum_{s'} P_o^\nu(\cdot \mid s') T(s' \mid b, a)$. Consider the joint distribution $\gamma^*$ over the space $\mathcal{S} \times \mathcal{S}$ given by $(s', s'_o)$ sampled as $s_o \sim T(\cdot|b, a), s'_o \sim P_o^\nu(\cdot|s')$. It is easy to check that $\gamma^*$ is a coupling, i.e., $\gamma^* \in \Gamma(P_o(\cdot|b, a), T(\cdot|b, a))$. We show this and for this we drop the dependency on $b, a$ for ease of notation. First, $\gamma^*(A, B) = \int_{A \times B} d\gamma^*(s', s'_o) = \int_A P_o^\nu(B|s') dT(s')$. Thus, $\gamma^*(A, \mathcal{S}) = \int_A dT(s') = T(A)$ and $\gamma^*(\mathcal{S}, B) = \int_\mathcal{S} P_o^\nu(B|s') dT(s') = P_o(B)$. Also, note that $||s' - s'_o||_\infty \leq \epsilon$ for $d$ as the infinity norm because of the bound of adversarial perturbation implicit in $P_o^\nu$. Then,

$$\begin{aligned} W_1(P_o(\cdot \mid b, a), T(\cdot \mid b, a)) &= \inf_{\gamma \in \Gamma(P_o(\cdot|b,a), T(\cdot|b,a))} \int ||s' - s'_o||_\infty d\gamma^*(s', s'_o) \\ &\leq \int ||s' - s'_o||_\infty d\gamma(s', s'_o) \\ &\leq \epsilon \end{aligned} \tag{7}$$

Combining Eq. 5 and Eq. 7 by triangle inequality we get

$$W_1(T(\cdot \mid s_o, a), P_o(\cdot \mid b, a)) \leq \xi + \epsilon$$

$\square$

The above results show that some basic structural properties are needed from the underlying system for bounding ACoE. One is that the value function should not change by a large amount due to small changes in state and another that the distribution of the next state should not be very different for two close by states. Clearly, an adversary can exploit systems that lack these properties.

*Proof of Proposition 4.1.* The proof is observed from the fact that C-ACoE can be viewed as an infinite horizon MDP with observations $s_o$ as states, immediate cost as $R(s_o, \pi(s_o)) - R(b(s_o), a)$, and transition to next state $s'_o$ described by $s'_o \sim \nu(s'), s' \sim T(\cdot \mid s, a)$. □

## B ADAPTATION FOR DQN

---

**Algorithm 2: $\delta$-DQN**

---

1 Initialize network $\delta_w$ with random weights $w$ and target network $\widehat{\delta}_{w^-}$ with weights $w^- = w$
2 Initialize network $Q_\theta$ with random weights $\theta$ and target network $\widehat{Q}_{\theta^-}$ with weights $\theta^- = \theta$
3 Initialize replay buffer $B$
4 Set robustness temperature $\lambda$
5 **for** *episode* $\in \{1, \ldots, M\}$ **do**
6     **for** $t = 0 \to H$ **do**
7         With prob. $1 - \epsilon$, select $a^t \in \arg\max_a Q_\theta(s_o^t, a) - \lambda \delta_w(s_o^t, a)$, else select $a^t$ at random
8         Sample $k$ states in $N(s_o)$, compute $b(s)$ for each $s \in N(s_o)$
9         Compute C-ACoE: $\delta_R = R(s_o^t, a^t) - \sum_{s \in N(s_o)} b(s) R(s, a^t)$
10         Execute action $a_t$, get observed state $s_o^{t+1}$, store transition $B = B \cup (s_o^t, s^t, s_o^{t+1}, \delta_R)$
11         Sample mini-batch $M \sim D$;
12         **for** *each* $(s_o^i, a^i, s_o^{i+1}, \delta_R^i)$ *in mini-batch $M$* **do**
13             Set target $y_i = \begin{cases} \delta_R^i, & \text{if episode terminates at step } i + 1 \\ \delta_R^i + \gamma \min_{a'} \delta_{w^-}(s_o^{i+1}, a'), & \text{otherwise} \end{cases}$
14             Set target $q_i = \begin{cases} R(s_o^t, a^t), & \text{if episode terminates at step } i + 1 \\ R(s_o^i, a^i) + \gamma \min_{a'} Q_{\theta^-}(s_o^{i+1}, a'), & \text{otherwise} \end{cases}$
15         Perform a gradient descent to update $w$ using loss: $\sum_{i=1}^{|M|} \left[ y_i - \delta_w(s_o^i, a^i) \right]^2$
16         Perform a gradient descent to update $\theta$ using loss: $\sum_{i=1}^{|M|} \left[ q_i - Q_\theta(s_o^i, a^i) \right]^2$
17     Every $K$ steps reset $w^- = w$ and $\theta^- = \theta$;

---

## C ESTIMATION OF BELIEF FOR CONTINUOUS STATE SPACE

**Lemma C.1.** *Assume $z(s) < B$ for some constant $B$. Consider $n$ uniformly random samples from $C$ stored in $N(s_o)$. Let $R$ and $\hat{R}$ be as defined above. Then, $(1/n) \sum_{s' \in N(s_o)} e^{z(s')}$ is an unbiased estimate of $(1/vol(C)) \int_{s \in C} e^{z(s)} ds$. There exists $n$ large enough so that $1 + \epsilon > \frac{(1/vol(C)) \int_{s \in C} e^{z(s)} ds}{(1/n) \sum_{s' \in N(s_o)} e^{z(s')}} > 1 - \epsilon$ with probability $1 - \delta$ for given small $\epsilon, \delta$. And then, $R(1 + \epsilon) > E[\hat{R}] > R(1 - \epsilon)$ with probability $1 - \delta$.*

*Proof.* Note that $E_{s' \sim U}[e^{z(s')}] = (1/vol(C)) \int_{s \in C} e^{z(s)} ds$, which gives us the first unbiasedness result. The second result comes from a straightforward application of Hoeffding's concentration

inequality where the bound $B$ is used. Then, we can see that

$$
\begin{aligned}
E[\hat{R}] &= \int_{s_1 \in C} \cdots \int_{s_n \in C} \frac{\sum_i R(s_i, a) e^{z(s_i)}}{\sum_i e^{z(s_i)}} u(s_1) \ldots u(s_n) ds_1 \ldots ds_n \\
&= \int_{s_1 \in C} \cdots \int_{s_n \in C} \frac{\sum_i R(s_i, a) e^{z(s_i)}}{\int_{s \in C} e^{z(s)} ds} \frac{\int_{s \in C} e^{z(s)} ds}{\sum_i e^{z(s_i)}} u(s_1) \ldots u(s_n) ds_1 \ldots ds_n \\
&\leq \frac{(1 + \epsilon) vol(C)}{n} \int_{s_1 \in C} \cdots \int_{s_n \in C} \frac{\sum_i R(s_i, a) e^{z(s_i)}}{\int_{s \in C} e^{z(s)} ds} u(s_1) \ldots u(s_n) ds_1 \ldots ds_n \\
&= \frac{(1 + \epsilon) vol(C)}{n} \sum_i \int_{s_i \in C} \frac{R(s_i, a) e^{z(s_i)}}{\int_{s \in C} e^{z(s)} ds} u(s_i) ds_i \\
&= \frac{(1 + \epsilon) vol(C)}{n} \times \frac{n}{vol(C)} \int_{s_i \in C} R(s_i, a) p(s_i) ds_i \\
&= (1 + \epsilon) R
\end{aligned}
$$

A similar argument holds for the lower bound, thereby, leading to the required result, □

## D DEFINING ACOE BELIEF METHODS WITH STATE HISTORIES

As mentioned in the paper, our methods are amenable to LSTM state histories as well, although empirically we find it to be not necessary (Table 9). Below, we define A2B and A3B when considering a state history of length 2.

**A2B:** Consider a time window of two with the current observation as $s_{o,1}$ and the previous observation as $s_{o,0}$.

$$
b(s_1, s_0) = \frac{e^{D_{KL}(\pi(s_1, s_0) || \pi(s_{o,1}, s_{o,0}))}}{\sum_{(s_1', s_0') \in N(s_{o,1}) \times N(s_{o,1})} e^{D_{KL}(\pi(s_1', s_0') || \pi(s_{o,1}, s_{o,0}))}}
$$

and

$$
b(s_1) = \sum_{s_0 \in N(s_{o,0})} b(s_1, s_0)
$$

For the initial timestep $s_{o,0}$ should be fixed to some constant, i.e. using the single-state A2B formula. This formulation does scale exponentially with the size of the neighborhoods, however we can scale down the previous state's neighborhood by considering a subset $s_0 \in N(s_{o,0})$ that had the highest belief.

**A3B**:

$$
b(s_1, s_0) = \frac{e^{z(s_1, s_0)}}{\sum_{(s_1', s_0') \in N(s_{o,1}) \times N(s_{o,1})} e^{z(s_1', s_0')}}
$$

and

$$
b(s_1) = \sum_{s_0 \in N(s_{o,0})} b(s_1, s_0) \ .
$$

Here,

$$
z(s_1, s_0) = \frac{D_{KL}(\pi(s_{o,1}, s_{o,0}) || \pi(s_1, s_0)}{D_{KL}(\pi(\nu(s_1), \nu(s_0)) || \pi(s_1, s_0))}
$$

## E ADDITIONAL EXPERIMENTAL RESULTS

We provide empirical investigations into a number of specifics that were cut from the main paper for space. Namely, fine-grained evaluations against long-horizon attack strategies in Figures 2, 3 and 4, and further empirical comparison to Protected-PPO (Liu et al., 2024). We also provide an extended version of the results tables in the main paper in Table 5 and 6 which include a few more baselines, namely CARRL (Everett et al., 2020), BCL (Wu and Vorobeychik, 2022), and CAR-DQN (Li et al., 2024).

### E.1 LONG-HORIZON ADVERSARIES

In prior works published before c. 2023, robust RL methods had been evaluated against myopic adversaries (i.e. adversaries give perturbations based on the current observation and victim policy, independent of future states and actions), and long-horizon adversarial actors were not considered. In more recent works PA-AD (Sun et al., 2023) is considered, however there are a variety of approaches each with distinct targeting strategies that can be evaluated. In our additional experiments, we include assessments of robust RL methods against the Strategically Timed attack (Lin et al., 2017), where the attacker computes the most effective attack intervals, and the Critical Point attack (Sun et al., 2020), in which the attacker delivers perturbations after computing the score reduction $N$ steps into the future.

We omit Protected-PPO from these granular long-horizon adversary experiments because these adversaries learn to attack a fixed victim policy at test time, and as the Protected-PPO method adapts over multiple episodes at test time, a fair comparative methodology is unclear. For worst-case PA-AD results with Protected-PPO, we refer to Table 8 and the PA-AD experiments table in the main paper.

### E.2 EMPIRICAL EVALUATIONS WITH PROTECTED-PPO

**Online Adaptations**: The most up-to-date robust RL method in this space is Protected-PPO (Liu et al., 2024), which computes a set of non-dominated policies during training. A key part of this method is the test time adaptation step in which a regret minimization algorithm (EXP3) with the set of policies is run for multiple rounds (each round is full policy episode) and the weights are updated *at test time* based on empirical performance against a fixed adversary, over $T = 800$ rounds of EXP3 ((Liu et al., 2024) reports 800, but we find the actual convergence to be faster in most environments). Because the evaluation setup for this method is quite different from all existing literature, we provide an empirical investigation into how the method performs under standard test setups as it is helpful to understand how it fits into the robust RL landscape.

The applications of interest for safe and robust RL such as autonomous vehicle or industrial control realistically do not accommodate any margin for error within one episode, let alone adaptation of a policy over multiple episodes.

To this end, we test the performance of Protected-PPO without any test time adaptation ($T = 1$, which denoted with † in the main paper) and with limited test time adaptation ($T = 10$). In Table 8, we find the unadapted policy performs poorly compared to the weakly-adapted counterpart, which is more uniformly robust. We also note that the weakly-adapted threshold of ($T = 10$) adaptation rounds doesn't improve performance uniformly across domains, as *Ant* and *Hopper* both become robust in that short time while *Walker* does not.

**LSTM History Length**: In Table 7, we also perform an investigation into the importance of an LSTM history for the Protected framework. We provide results for a Protected-PPO model using only linear hidden layers, labeled Protected$^{H=1}$. We find that the state history is quite integral to the performance of the method, which functions as the belief about the adversary for the method. This supports the ideas that the partially-observable nature of adversarial RL is the main challenge and must be addressed.

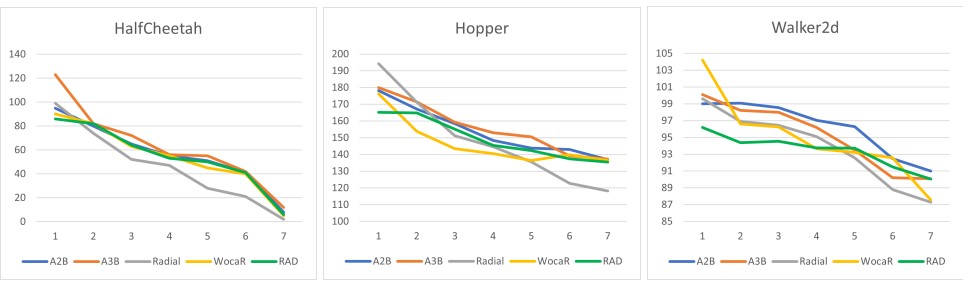

Figure 2: Robust agents vs. a Critical Point strategic adversary (Sun et al., 2020) with increasing search sizes.

Table 5: Experimental results versus myopic adversaries. Most robust scores are in **bold**. Methods are evaluated on DQN implementations in Atari and Highway, with adversarial perturbation bounds permitted as $\epsilon$=0.1 for PGD, and 0.15 for MAD. *CAR-DQN results are reported directly from their publication, which only uses PGD $\epsilon$=0.02.

| Method | Unperturbed | MAD | PGD | Unperturbed | MAD | PGD |
|---|---|---|---|---|---|---|
| | | highway-fast-v0 | | | merge-v0 | |
| PPO | 24.8±5.42 | 13.63±19.85 | 15.21±16.1 | 14.94±0.01 | 10.2±0.02 | 10.42±0.95 |
| CARRL | 24.4±1.10 | 4.86±15.4 | 12.43±3.4 | 12.6±0.01 | 12.6±0.01 | 12.02±0.01 |
| RADIAL | 28.55±0.01 | 2.42±1.3 | 14.97±3.1 | 14.86±0.01 | 11.29±0.01 | 11.04±0.91 |
| WocaR | 21.49±0.01 | 6.15±0.3 | 6.19±0.4 | 14.91±0.04 | 12.01±0.28 | 11.71±0.21 |
| RAD | 21.01±0.01 | 20.59±4.1 | 20.02±0.01 | 13.91±0.01 | 13.90±0.01 | 11.72±0.01 |
| A2B | 24.8±0.01 | 23.11±0.01 | 20.8±12.6 | 14.91±0.01 | 14.23±0.8 | 12.92±0.13 |
| A3B | 23.8±0.01 | **23.21±0.01** | **22.61±14.1** | 14.91±0.17 | **14.88±0.17** | **14.89±0.17** |
| | | roundabout-v0 | | | intersection-v0 | |
| PPO | 10.33±0.40 | 7.41±0.69 | 3.92±1.35 | 9.26±7.6 | 3.62±11.63 | 6.75±12.93 |
| CARRL | 9.75±0.01 | 9.75±0.01 | 5.92±0.12 | 8.0±0 | 7.5±0 | 9.0±0.1 |
| RADIAL | 10.29±0.01 | 5.33±0.01 | 8.77±2.4 | 10.0±0 | 2.4±5.1 | 9.61±0.1 |
| WocaR | 6.75±2.5 | 6.05±0.14 | 6.48±2.7 | 10.0±0.05 | 9.47±0.3 | 3.26±0.4 |
| RAD | 9.22±0.3 | 8.98±0.3 | 9.11±0.3 | 9.85±1.2 | 9.71±2.3 | 9.62±0.1 |
| A2B | 10.5±0.0 | 10.1±0.1 | 10.0±0.5 | 10.0±0 | **10.0±0** | **9.88±0.12** |
| A3B | 10.5±0.01 | **10.33±0.01** | **10.18±2.1** | 10.0±0 | 9.68±0 | **9.88±0.1** |
| Method | Unperturbed | MAD | PGD | Unperturbed | MAD | PGD |
| | | Pong | | | Freeway | |
| PPO | 21.0±0 | -20.0±0.07 | -19.0±1.0 | 29 ± 3.0 | 4 ± 2.31 | 2±2.0 |
| CARRL | 13.0 ±1.2 | 11.0±0.010 | 6.0±1.2 | 18.5±0.0 | 19.1 ±1.20 | 15.4±0.22 |
| BCL | 21± 0 | – | **21± 0** | 34.0 ± 0 | – | 21.2± 0.5 |
| CAR-DQN* | 21± 0 | – | **21± 0** | 34.0 ± 0 | – | 33.7 ± 0.1 |
| RADIAL | 21.0±0 | 11.0±2.9 | **21.0± 0.01** | 33.2±0.19 | 29.0±1.1 | 24.0±0.10 |
| WocaR | 21.0±0 | 18.7 ±0.10 | 20.0 ± 0.21 | 31.2±0.41 | 19.8±3.81 | 28.1±3.24 |
| RAD | 21.0±0 | 14.0 ± 0.04 | 14.0 ± 2.40 | 33.2±0.18 | 30.0±0.23 | 27.7±0.2 |
| A2B | 21.0±0 | 20.1±0.04 | **21.0±0.01** | 33.2±0.18 | 30.1±0.43 | 30.8±1.51 |
| A3B | 21.0±0 | **20.8±0.7** | **21.0±0.01** | 33.2±0.18 | **31.0±0.87** | **31.1±1** |
| | | BankHeist | | | RoadRunner | |
| PPO | 1350±0.1 | 680±419 | 0±116 | 42970±210 | 18309±485 | 10003±521 |
| CARRL | 849±0 | 830±32 | 790±110 | 26510±20 | 24480±200 | 22100±370 |
| BCL | 1215 ± 8.4 | – | 894.1± 9.2 | 42490±1309 | – | 23291±1121 |
| CAR-DQN* | 1349 ± 3 | – | **1347±3.6** | 49700±1015 | – | **43286±801** |
| RADIAL | 1349±0 | 997±3 | 1130±6 | 44501±1360 | 23119±1100 | 24300±1315 |
| WocaR | 1220±0 | 1207±39 | 1154±94 | 44156±2270 | 25570±390 | 12750±405 |
| RAD | 1340±0 | 1170±42 | 1211±56 | 42900±1020 | 29090±440 | 27150±505 |
| A2B | 1350±0 | **1230±42** | 1240±56 | 44050±1020 | 38205±440 | 40015±505 |
| A3B | 1350±0 | **1230±12** | **1250±30** | 44290±1250 | **41001±610** | **42645±458** |
| Method | Unperturbed | MAD | PGD | Unperturbed | MAD | PGD |

Table 6: Experimental results versus myopic adversaries. Most robust scores are in **bold**. Methods are evaluated on PPO implementations in Mujoco, with adversarial perturbation bounds permitted as $\epsilon$=0.1 for PGD, and 0.15 for MAD. Protected-PPO is grayed out due to differences in evaluation methodology as outlined in the main paper. For fine-grained comparisons, see Tables 7 and 8.

| | Hopper | | | Walker2d | | |
|---|---|---|---|---|---|---|
| PPO | 4128 ± 56 | 1110±32 | 128±105 | 5002 ± 20 | 680±1570 | 730±262 |
| RADIAL | 3737±75 | 2401±13 | 3070±31 | 5251±10 | 3895±128 | 3480±3.1 |
| WocaR | 3136±463 | 1510 ± 519 | 2647 ±310 | 4594±974 | 3928±1305 | 3944±508 |
| Protected | 3652±108 | 2512±392 | 2221± 775 | 6319±31 | 5148±1416 | 4720± 1508 |
| RAD | 3473±23 | 2783±325 | 3110±30 | 4743±78 | 3922±426 | 4136±639 |
| A2B | 3710±11 | 3240±41 | 3299±28 | 4760±61 | 4636±87 | 4708±184 |
| A3B | 3766±23 | **3370±275** | **3465±17** | 5341±60 | **5025±94** | **5292±231** |
| | HalfCheetah | | | Ant | | |
| PPO | 5794 ± 12 | 1491±20 | -27±1288 | 5620±29 | 1288±491 | 1844±330 |
| RADIAL | 4724±76 | 4008±450 | 3911±129 | 5841±34 | 3210±380 | 3821±121 |
| WocaR | 5220±112 | 3530±458 | 3475±610 | 5421±92 | 3520±155 | 4004±98 |
| Protected | 7095±88 | 4792±1480 | 4680±1203 | 5769±290 | 4440±1053 | 4228± 484 |
| RAD | 4426±54 | 4240±4 | 4022±851 | 4780±10 | 3647±32 | 3921±74 |
| A2B | 5192 ±56 | 4855± 120 | 4722±33 | 5511±13 | 3824±218 | 4102±315 |
| A3B | 5538±20 | **4986±41** | **5110±22** | 5580±41 | **4071±242** | **4418±290** |

Table 7: Comparison to the *Protected* framework Liu et al. (2024) with a history of only one state. Here, we demonstrate superior robust performance when information is limited.

| Method | Unperturbed | MAD | Unperturbed | MAD |
|---|---|---|---|---|
| | Hopper | | Walker2d | |
| PPO | 4128 ± 56 | 1110±32 | 5002 ± 20 | 680±1570 |
| WocaR | 3136±463 | 1510 ± 519 | 4594±974 | 3928±1305 |
| Protected$^{H=1}$ | 2451±81 | 2198±233 | 3509±32 | 3410±41 |
| A2B | 3710±11 | 3240±41 | 4760±61 | 4636±87 |
| A3B | 3766±23 | **3370±275** | 5341±60 | **5025±94** |
| | HalfCheetah | | Ant | |
| PPO | 5794 ± 12 | 1491±20 | 5620±29 | 1288±491 |
| WocaR | 5220±112 | 3530±458 | 5421±92 | 3520±155 |
| Protected$^{H=1}$ | 3210±18 | 2241±392 | 3997±285 | 2331±277 |
| A2B | 5192 ±56 | 4855± 120 | 5511±13 | 3824±218 |
| A3B | 5538±20 | **4986±41** | 5580±41 | **4071±242** |

Table 8: Comparison to the *Protected* framework Liu et al. (2024) with zero test time adaptation (labelled $T = 1$), for an apples-to-apples evaluation comparison to existing baselines. Without the online adaptation part of the Protected framework, we find robust performance (i.e. low drop in score) but not high nominal scores. $T = 10$ allows Protected to adapt for limited number of rounds.

| Method | Unperturbed | MAD | PA-AD | Unperturbed | MAD | PA-AD |
|---|---|---|---|---|---|---|
| | | Hopper | | | Walker2d | |
| Protected$^{T=1}$ | 3573±81 | 2398±665 | 2210±385 | 5019 ± 87 | 3887 ± 492 | 4480 ± 492 |
| Protected$^{T=10}$ | 3691±81 | 3314±391 | 3221±222 | 6001 ± 24 | 3410 ± 558 | 5520 ± 31 |
| A2B | 3710±11 | 3240±41 | 2441 ±31 | 4760±61 | 4636±87 | 3997±214 |
| A3B | 3766±23 | **3370±275** | 2580±92 | 5341±60 | **5025±94** | 4931±166 |
| | | HalfCheetah | | | Ant | |
| Protected$^{T=1}$ | 4777±360 | 3997±285 | 2331±277 | 4620±32 | 4264±166 | 3103± 96 |
| Protected$^{T=10}$ | 5722±58 | 5296±411 | 4522±450 | 4747±59 | 4688±201 | 4186±8 |
| A2B | 5192 ±56 | 4855± 120 | 4393±79 | 5511±13 | 3824±218 | 2821 ± 312 |
| A3B | 5538±20 | **4986±41** | 4478±67 | 5580±41 | **4071±242** | 3205±275 |

Table 9: Empirical analysis between single-state ACoE and LSTM-ACoE on discrete-action domains (top, *highway-env*) and contiuous-action domains (bottom, *Mujoco*). Single-state PPO included as a point of reference.

| Method | Unperturbed | MAD | PGD | Unperturbed | MAD | PGD |
|---|---|---|---|---|---|---|
| | | highway-fast-v0 | | | merge-v0 | |
| PPO | 28.8±5.42 | 13.63±19.85 | 15.21±16.1 | 14.94±0.01 | 10.2±0.02 | 10.42±0.95 |
| A3B | 25.8±0.01 | 24.21±0.01 | 22.61±14.1 | 14.91±0.17 | 14.88±0.17 | 14.89±0.17 |
| A3B-LSTM | 28.8±0.01 | 25.21±0.01 | 23.03±14.1 | 14.96±0.1 | 14.88±0.1 | 14.90±0.15 |
| | | Halfcheetah | | | Hopper | |
| PPO | 5794±12 | 1491±20 | 5620±29 | 4128 ± 56 | 1110±32 | 5002 ± 20 |
| A3B | 5538±20 | 4986±41 | 5110±22 | 3766±23 | 3370±275 | 3465±17 |
| A3B-LSTM | 5641±34 | 5002±67 | 5171±88 | 3729±45 | 3411±137 | 3453± 21 |

Table 10: Ablation study: relaxing test-time attacker constraint $\epsilon$ shows lower score degradation in ACoE agents than SOTA Protected agents.

| Method | MAD attack $\epsilon$ = 0.15 | = 0.175 | = 0.2 | = 0.3 |
|---|---|---|---|---|
| Halfcheetah | | | | |
| A3B | 4986 ± 41 | 5008 ± 259 | 4907 ± 200 | 3896 ± 1477 |
| Protected$^{T=10}$ | 4551 ± 843 | 4391± 729 | 3855 ± 1718 | 2410 ± 1880 |
| Hopper | | | | |
| A3B | 3512±112 | 3470± 66 | 3367± 208 | 3023 ± 348 |
| Protected$^{T=10}$ | 3484±73 | 3312±119 | 3290± 249 | 2705±396 |

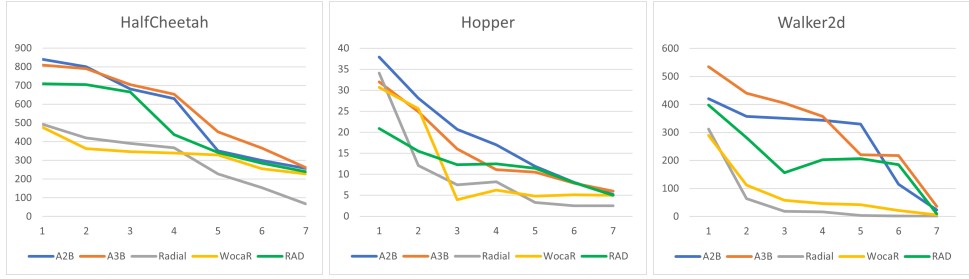

Figure 3: Robust agents vs. a Strategically Timed Attack adversary (Lin et al., 2017), as the length of perturbation increases. We find that as the level of strategy increases from long-horizon attackers, C-ACoE minimization improves robust performance, relative to other methods.

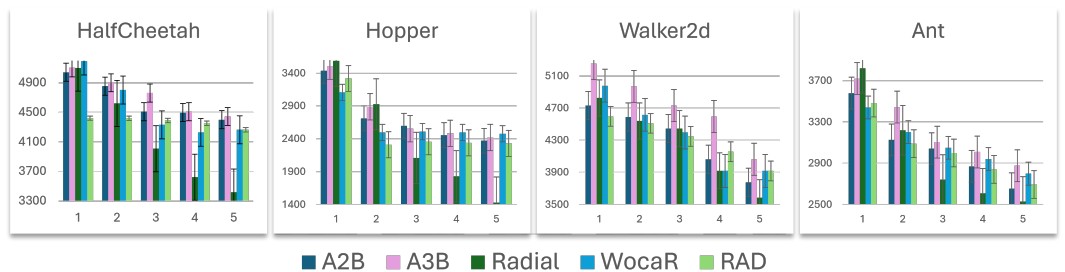

Figure 4: Robust agents vs. a PA-AD attacker (Sun et al., 2023), as the optimality of the attacker policy increases. To represent levels of optimality, we save PA-AD model weights at 5 evenly distributed points across the training epochs. We find that as the level of strategy increases from long-horizon attackers, C-ACoE minimization achieves more robust performance, relative to other methods.

### E.3  ABLATION STUDIES ON HYPERPARAMETERS

In Tables 10 and 11, we examine sensitivities to different training parameters used in the ACoE framework. We train several different ACoE models in Mujoco-halfcheetah, varying the denoted parameters. We determine that while the robustness-sensitivity parameter $\lambda$ does have some effect on the robustness/value tradeoff, it is not sensitive to small changes. We find no significant impact of the neighborhood sample size on performance, due to the use of Softmax which favors extreme values.

In Table 9, we observe the improvements made to ACoE when including a two-state LSTM history as the Protected framework uses, and find that while the performance does marginally increase the unperturbed score. However, the trade-off is expensive, as applying ACoE to each state in a history is combinatorially complex.

Table 11: Ablation study: training parameters. We train several different ACoE models in Mujoco-halfcheetah, varying the denoted parameters. We determine that the robustness-sensitivity parameter $\lambda$ is not sensitive to small changes. We find no significant impact of the neighborhood sample size on performance.

| $\lambda$ **value:** | 0.1 | 0.19 | 0.2 | 0.21 | 0.3 | 0.5 |
|---|---|---|---|---|---|---|
| **ACoE unperturbed:** | $5620 \pm 40$ | $5578 \pm 38$ | $5538 \pm 20$ | $5557 \pm 19$ | $4994 \pm 12$ | $4286 \pm 23$ |
| **ACoE vs. MAD:** | $4897 \pm 62$ | $4971 \pm 47$ | $4986 \pm 41$ | $5002 \pm 48$ | $4731 \pm 28$ | $4021 \pm 30$ |
| **# Nbhd samples:** | 2 | | 10 | | 20 | |
| **ACoE unperturbed:** | $5521 \pm 23$ | | $5528 \pm 20$ | | $5535 \pm 13$ | |
| **ACoE vs. MAD:** | $4981 \pm 35$ | | $4986 \pm 41$ | | $4990 \pm 38$ | |

# F  SUBJECTIVE ANALYSIS

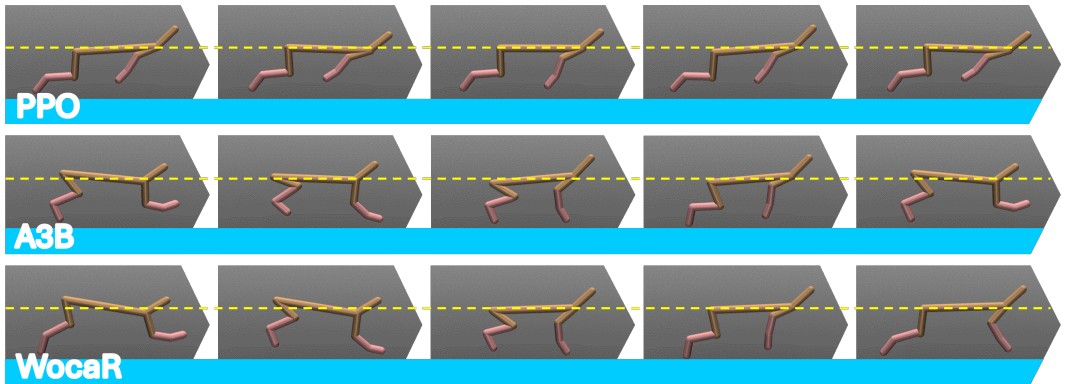

Figure 5: Last 5 frames of PPO, A3B, and WocaR agents (top to bottom), on MuJoCo-*HalfCheetah*. PPO deviates the least from the dashed center-mass line, and has the least balanced gait. WocaR has arguably the most stable posture when noting the faster front leg recovery of A3B, but our empirical results suggest optimizing maximum stability is not always necessary. Full GIFs: tinyurl.com/a3b-gif

In Figure 5, we show the visual differences frame-by-frame between PPO, A3B, and Wocar-trained models. A3B and Wocar agents exhibit visually similar behavior, which are distinctly more stable than the PPO-learned behavior. Subjectively speaking, the robust behavior is more realistic and accurately depicts how one would expect the agent to move, while the PPO behavior is more of an exploitation of the MuJoCo physics engine than a realistic behavior. Under adversary this becomes relevant: the niche value-optimal exploitative movement of the PPO agent is in turn exploited by an adversary, while the robust models can retain their stability.

# G  TRAINING DETAILS AND HYPERPARAMETERS

## G.1  MODEL ARCHITECTURE

Our DQN and PPO models follow settings common to the current lineage of robust RL work (SA-MDP, Radial, WocaR, RAD). For C-ACoE estimator functions, we use two 64x hidden layers with a single linear output layer, congruent to the CCER estimator in RAD and Worst-value estimator in WocaR. For Atari image domains, we use a convolutional layer with an 8x8 kernel, stride of 4 and 32 channels, a convolutional layer with a 4x4 kernel, stride of 2 and 64 channels, and a final convolutional layer with a 3x3 kernel, stride of 1 and 64 channels. Each layer is followed by a ReLU activation, and finally feeds into a fully connected output.

The LSTM models use a 64x64 hidden layer size with linear layers for input and output.

## G.2  TRAINING HYPERPARAMETERS

We train our methods for 900 episodes for all MuJoCo environments, using an annealed (Adam) learning rate of $0.005$. The robustness hyperparameter $\lambda$ is set to $0.2$ for all of our models, which is the same as the robustness hyperparameters found in prior works Oikarinen et al. (2021); Liang et al. (2022); Belaire et al. (2024); Zhang et al. (2020). The attack neighborhood sample size is set to 10, and the training attack neighborhood radius is set to $\epsilon = 0.1$, both tuned from sets in the range $\pm 100\%$. All other hyperparameters are the same as those used in Liang et al. (2022), which is open-sourced at https://github.com/umd-huang-lab/WocaR-RL.

## G.3  HARDWARE

We train our linear models on an NVIDIA Tesla V100 with 16gb of memory, and LSTM models on an NVIDIA L40 32gb GPU.

