# OpenReview forum: "On Minimizing Adversarial Counterfactual Error in Adversarial Reinforcement Learning"
_ICLR.cc/2025/Conference — ICLR 2025 Poster_

### Official Review · Reviewer_X3rZ · 2024-10-28

**Soundness:** 2
**Presentation:** 2
**Contribution:** 3
**Rating:** 6
**Confidence:** 2

**Summary:**

This paper introduces the Adversarial Counterfactual Error (ACoE) as a metric for quantifying robustness in adversarial deep reinforcement learning (DRL). Current methods for adversarial robustness often struggle to balance performance and robustness, especially in cases of partial observability due to adversarial noise. The authors hypothesise that explicitly accounting for partial observability in adversarial settings can lead to more balanced results. To address this, they propose Cumulative-ACoE (C-ACoE), a scalable surrogate objective designed to improve model-free DRL methods. C-ACoE allows for simultaneous robustness and performance improvements achieved through adaptations of popular DRL algorithms such as PPO and DQN. Their experimental results on various benchmarks (MuJoCo, Atari, and Highway) show that C-ACoE methods outperform current state-of-the-art approaches in adversarial environments.

**Strengths:**

The paper proposes an original approach to managing adversarial robustness in DRL by framing it as a partial observability problem. This novel perspective on adversarially-induced partial observability, operationalized by ACoE and its scalable form C-ACoE, is a significant theoretical contribution. With ACoE as the objective, the methods A2B and A3B show superior performance compared to baselines.

**Weaknesses:**

- The Abstract is overly long.
- Missing $)$ in Eq. 2.
- It is unclear to me how C-ACoE $delta$-network is learned, and how it's aligned with the theory especially Eq. (2). Specifically, in the paper the authors claim that $delta$ is no longer dependent on $b$, then what's the use of $b(s)$ in A2B and A3B? And if $b$ is not needed then in Eq. (2) RHS the second term still has $b$, then how do you learn C-ACoE correctly? The authors should put more details about C-ACoE in algorithm 1, and omit the common knowledge like GAE and PPO-clipped.

**Questions:**

1. Do we care about unperturbed performance since basically all real-world applications have states being perturbed. Can you list any applications?
2. Both A2B and A3B require KL divergence between policies, which should be computational costly. Did the authors address this anywhere?
3. Can you provide any intuition about algorithmically, why this method can have such good empirical performance?

---

> ### Author Response · Authors · 2024-11-19
>
> We sincerely thank the reviewer for their valuable comments and insights. Below, we address the points raised in the review.
>
> 1) **Do we care about unperturbed performance since basically all real-world applications have states being perturbed?**
> To answer this, we should highlight the difference between adversarial perturbations and non-adversarial or stochastic perturbations. The latter case can be present in the training environment and is accounted for in the expected value of a policy. To this end, PPO is generally sufficient. We include the adversarially unperturbed case (which some call the "natural" reward) in our experiments to showcase the trade-off between adversarially-unperturbed and adversarially-perturbed rewards for each algorithm.
>
> 2) **Both A2B and A3B require KL divergence between policies, which should be computational costly.**
> We would like to clarify a misunderstanding here: our method computes the KL-Divergence between two probability distributions over actions in a state and not the entire policies, thus that step is not computationally costly.
>
> 3) **Can you provide any intuition about algorithmically, why this method can have such good empirical performance?**
> Intuitively, the identification of the adversarial problem as a POMDP allows our methods to assign appropriately cautious actions by forming beliefs about the true state. Our belief construction puts more probability mass on states that are most likely given adversarial intent and reduces unnecessary focus on unlikely perturbations.
>
> 4) **the authors claim that $\delta$ is no longer dependent on $b$, what's the purpose of $b(s)$ in A2B and A3B?**
> Regarding Eq. 2: We state that C-ACoE $\delta$ is a function of $s_o$ only, since just above that equation we state that $b(s_o)$ is also a function of $s_o$ only, that is, $\delta(s_o, b(s_o))$ can be viewed as a function of $s_o$ only as the second input $b$ can be constructed completely from $s_o$. This is also why in the RHS of Eq. 2 we can write $b(s_o)$, that is, the belief as a function of input $s_o$. These belief functions are later given by A2B/A3B.
>
> _We will update the PDF with our new results and typo fixes after performing all experiments asked by all reviewers._

---

### Official Review · Reviewer_Ui4m · 2024-11-01

**Soundness:** 3
**Presentation:** 4
**Contribution:** 3
**Rating:** 6
**Confidence:** 3

**Summary:**

This paper focus on the adversarial defense in the context of Deep Reinforcement Learning (DRL). Specifically, this paper introduces Adversarial Counterfactual Error (ACoE) defined based on the beliefs (uncertainty) about the underlying true state of the target model, thus balancing value optimization for improved robustness. In addition to extensive experimental evaluations, the authors also provide systematic analyses in the theoretical part to justify the effectiveness and generalization ability of the proposed method.

**Strengths:**

1. The proposed method is novel with a clear motivation. It's also scalable for model-free environments.
2. Extensive experiments and analyses are conducted to demonstrate the efficacy of the proposed method.
3. Theoretical analyses are detailed and solid.

**Weaknesses:**

1. This paper has limited (experimental) justifications for surrogate belief construction choices. Comparisons with alternative constructions would be better to support the motivation.
2. The method's adaptability to diverse or higher-dimensional continuous state spaces is underexplored. Instead of theoretical analyses, further explanations or experimental results would be preferred.
3. I feel the performance of the proposed C-ACoE would be likely sensitive to hyper- parameters. Further analyses are required.

**Questions:**

1. Is it possible to provide an analysis of the trade-off between natural performance and adversarial robustness?
2. Is it possible to extend the proposed method to high-dimensional continuous state spaces?
3. The authors can try to run sensitivity analyses on key hyperparameters, such as the
robustness-weight factor \lambda and neighbourhood size

---

> ### Author Response · Authors · 2024-11-19
>
> We sincerely thank the reviewer for their valuable comments and insights. Below, we address the points raised in the review.
>
> 1) **Analysis of the trade-off between natural performance and adversarial robustness**:
>  To check the tradeoff between natural and robust performance, we can examine the role of the $\lambda$ parameter, which dictates how value-seeking or safety-seeking the agent is. This term is also present in all baselines we compare to with the same value, $0.2$, in all domains. Below we provide a sensitivity analysis for this parameter in Table 3.1 for A3B. However, it would take some time to retrain each method for a full comparison between all baselines.
>
> 2) **Extension to high-dimensionality continuous domains**:
> We request the reviewer to refer to lines 365-373 on how we handle continuous domains.; We also note that the Atari domains have images as inputs, which are considered continuous data points and are extremely high dimensional (86*86*3*255).
>
> 3) **Sensitivity analyses on key hyperparameters**:
> Please see Tables 3.1 and 3.2 below for experiments on $\lambda$ and neighborhood size. We observe that for $\lambda$ close to 0.2, the performance is quite stable. The same goes for neighborhood sample size.
>
> Table 3.1 $\lambda$ variations in Halfcheetah
> |$\lambda$ value | unperturbed | MAD attack |
> |-|-|-|
> |0.1 | 5620 +- 40 | 4897 +- 62 |
> |0.19 | 5578 +- 38 | 4971 +- 47 |
> |0.2 | 5538 +- 20 | 4986 +- 41 |
> |0.21 | 5557 +- 19 | 5002 +- 48
> |0.3 | 4994 +- 12 | 4731 +- 28 |
> |0.5 | 4286 +- 23 | 4021 +- 30 |
>
>
> Table 3.2 Neighborhood Sample Size Variations in Halfcheetah
> |Neighborhood Size | Unperturbed | MAD Attack |
> |-|-|-|
> |2 samples | 5521 +- 23 | 4981 +- 35 |
> |10 samples | 5538 +- 20 | 4986 +- 41 |
> |20 samples | 5535 +- 13 | 4990 +- 38 |
>
>
> 4) **Surrogate belief construction choices.**:
> We propose two belief alternatives, A2B and A3B, that work in an unperturbed training environment. As far as we know, we are the first to suggest such surrogate beliefs, and we are not aware of other alternatives in the literature. We would be happy to compare to other belief constructions that the reviewer has in mind.
>
> _We will update the PDF with our new results and typo fixes after performing all experiments asked by all reviewers._

---

### Official Review · Reviewer_xpL3 · 2024-11-02

**Soundness:** 3
**Presentation:** 3
**Contribution:** 3
**Rating:** 6
**Confidence:** 4

**Summary:**

This paper proposes a novel method for adversarial reinforcement learning which balances between natural policy value and adversarial robustness. By revisiting the relation between policy value without perturbation and with perturbation, the paper proposes a new objective that minimizes the controllable difference of the return between attacked policy and unattacked policy. Based on the theoretical insights, the paper introduces a model-free adversarial training method with two algorithms, A2B and A3B, to estimate the belief states. Experiments on variaous environments and attack methods show that the proposed methods are effective, achieveing better balance between clean reward and attacked rewards.

**Strengths:**

- This paper proposes an interesting idea of balancing policy unperturbed value and perturbed value through the optimization of Adversarial Counterfactual Error. Different from prior work which considers the worst-case performance, the proposed method estimates the attacked value via belief state estimation.
- The paper is in general well-written and easy to follow.
- The paper presents experiments in several different scenarios and shows better performance than SOTA baselines. The method can be adapted for both PPO and DQN, both of which achieve good empirical results.

**Weaknesses:**

Both A2B and A3B rely on some knowledge of the attacker, such as attack radius and attack perference. Can the authors provide more justification on whether this is a realistic assumption? For the experiments, how does the attack radius and surrogate attack set for different attack scenarios? If the surrogate attack radius or attack preference is very different from the actual attack, how will the attack performance drop compared to baselines which consider the worse-case attack?

**Questions:**

See my questions in the weakness section.

---

> ### Author Response · Authors · 2024-11-19
>
> We sincerely thank the reviewer for their valuable comments and insights. Below, we address the points raised in the review.
>
> 1) **Both A2B and A3B rely on some knowledge of the attacker, such as attack radius and attack preference. Can the authors provide more justification on whether this is a realistic assumption?**
>
> To answer the reviewer’s question in two parts:
>
> 1a) the attack radius $\epsilon$ is often assumed in literature to be a bound outside which an attack is considered "detected" and subsequently disregarded or heuristically intervened upon. This is an assumption utilized in most adversarial ML and adversarial RL research. Utilizing unbounded $\epsilon$ would correspond empirically to converting a stop sign to a closed road and hence is not realistic/justified.
>
> 1b) In our paper, we assume an adversary is operating to reduce the reward of the victim agent. We find this to be an intuitive formulation, as this fits the widely followed zero-sum formulation of adversarial RL.
>
> 2) **How are attack parameters chosen in different scenarios?**
> We use the same attack radius for all environments $\epsilon=0.1$ (which is the same as prior methods [1,2]) in training, and $\epsilon\geq 0.15$ in attack testing. The surrogate attack we use is PGD for all training, chosen for consistency and theoretical soundness.
>
> 3) **How does performance change when the surrogate and actual attack constraints are different?**
> If the attacker is more constrained than expected (in terms of $\epsilon$), the performance of any robust method can be expected to improve. On the other hand, if the attacker is less constrained, our method experiences less performance degradation than SOTA (see Table 2.2). Regarding different attacker preferences, we can refer to Table 2.1.
>
> We also provide a quick comparison to an adversary that maximizes the victim reward as an example of a different opponent preference in Table 2.1. We observe that the performance of both our approach and the existing SOTA approach improves with such an adversary in place.
>
>
> _[1] Yongyuan Liang, Yanchao Sun, Ruijie Zheng, and Furong Huang. Efficient adversarial training without attacking: Worst-case-aware robust reinforcement learning. 2022_
>
> _[2] Tuomas Oikarinen,Wang Zhang,Alexandre Megretski,Luca Daniel,and Tsui-Wei Weng. Robust deep reinforcement learning through adversarial loss. 2021_
>
> Table 2.1 Alternate attacker preference example
> |Algorithm | Natural Reward | Max-Reward “Adversary” (halfcheetah) |
> |-|-|-|
> |PPO | 5794 +- 12 | 5851 +- 301 |
> |A3B | 5538 +- 20 | 5610 +- 51 |
> |PROTECTED | 4777 +- 360 | 4819 +- 942|
>
>
>
> Table 2.2 Attacker constraint analysis
> |Algorithm | $\epsilon$=0.15 | =0.175 | =0.2 | =0.3 |
> |---|---|---|---|---|
> |A3B | 4986 +- 41 | 5008 +- 259 | 4907 +- 200 | 3896 +- 1477 |
> |PROTECTED | 4551 +- 843 | 4391+- 729 | 3855 +- 1718 | 2410 +- 1880|
>
>
>
> _We will update the PDF with our new results and typo fixes after performing all experiments asked by all reviewers._

---

> > ### Comment · Reviewer_xpL3 · 2024-11-25
> > **Thank you for the response**
> >
> > Thank you for the detailed response and experiments. My major concerns are addressed.

---

### Official Review · Reviewer_BY2g · 2024-11-05

**Soundness:** 3
**Presentation:** 4
**Contribution:** 3
**Rating:** 6
**Confidence:** 3

**Summary:**

The paper introduced a new perspective into adversarial RL problems by viewing the state-perturbing adversary as inducing a POMDP. The authors then proposed the concept of Adversarial Counterfactual Error (ACoE) and developed a surrogate of it for applicability in general RL problems. Extensive experiments in various environments demonstrate the effectiveness of the proposed ACoE objective and the practical algorithms.

**Strengths:**

1. The newly introduced objective ACoE provides a theoretically grounded approach to handling the POMDP induced by state-perturbing adversaries. This is novel and a valuable contribution to the adversarial RL literature.
2. The evaluation is comprehensive. The authors experimented with various RL environments and considered different types of attacks, which consistently demonstrated the effectiveness of the proposed approach.
3. The paper is overall well-written and pleasant to read. The authors have done a great job setting up the background, comparing with the literature, motivating the new objective and algorithm on top of that.

**Weaknesses:**

I overall enjoy reading the paper and only have a few minor questions; see "Questions" section.

**Questions:**

1. The authors "restrict solutions to the set of policies that depend just on the current observation" (line 302). Can the authors discuss the trade-off here, i.e., how much benefit can there be if we consider using policies that can depend on the history, and what's the downside of the increased complexity (e.g. computational cost).
2. It's encouraging to see that the proposed approach performs well even against long-horizon adversaries, given that the design of the brief construction algorithms only involve the consideration for the myopic adversary. It would be interesting to see whether A3C can be hacked and to which extent--suppose we have a long-horizon adversary who's aware of the A3C heuristic (an adaptive attacker), how robust is the proposed algorithm in this case.
3. The authors focused on the state-perturbing adversary in this work. Can the authors discuss the extension of the framework to other types of RL adversary?

Minor

- Line 362: should be $\pi(s_1)$ instead of $\pi(s_1')$.

---

> ### Author Response · Authors · 2024-11-19
>
> We sincerely thank the reviewer for their valuable comments and insights. Below, we address the points raised in the review.
>
> 1) **Tradeoffs for using a history**:
> Including a history can give our algorithm a more informed estimate of the probability of the true state given observations. In Appendix D of the submission, we derive variants of A2B and A3B that work for a history of two observations. As can be seen from that derivation, if the past observations are assumed to be perturbations of true states as well, the belief computation scales combinatorially with the length of the history. On the other hand, if the history states are taken as ground truth, there is no change in computational complexity. In either case, there is no change in complexity at run time (a forward call of the network). Empirically, adding a one-state LSTM history to our model increases training time on Mujoco-Halfcheetah from $\sim 800$ episodes (4.5 hours) to $>3000$ episodes (16 hours) to obtain sufficient samples for the belief. However, there is little tangible benefit in doing so, as seen in Appendix Table 9 and Table 1.1 below.
>
> Table 1.1: A3B-LSTM on HalfCheetah
> |Algorithm | Unperturbed | MAD attack|
> |-------------|--------------|--------------------|
> |A3B | 5538 +- 20 | 4986 +- 41|
> |A3B-LSTM | 5641 +- 34 | 5002 +- 67|
>
>
> 2) **Algorithm-adaptive attackers**:
> Our approach A3C performs well against PA-AD adversaries and PA-AD attacks work in the white-box setting where the victim policy network is available to the adversary. As far as we know, we have not come across an attack where the adversary exploits knowledge of the underlying algorithm (more than white-box; open-box?). We believe that this would make for exciting future work, not only in this setting but in general adversarial machine learning as well.
>
> 3) **Other RL Adversaries**:
> In the space of online RL, there are two broad formulations of attacks: perturbations to observations, and perturbations to the environment (simulator) directly, such as perturbations to the transition function. Our method targets the first case, as the central tenet of ACoE is recognizing and addressing the mismatch between the observed and true state, i.e., the adversary does not modify the simulator directly. As the adversary can directly modify the simulator in the second case, our method isn’t directly applicable to such attacks. However, in a similar vein to our work, there is a possible approach of forming beliefs about the adversary’s modification of the simulator and acting accordingly in the context of environment attacks.
>
> _We will update the PDF with our new results and typo fixes after performing all experiments asked by all reviewers._

---

### Author Response · Authors · 2024-11-25
**Thanks to the Reviewers**

We sincerely appreciate the time you've dedicated to reviewing our work. Before the end of the discussion phase, we would like to check for any remaining critiques, comments, or questions, which we hope have been thoroughly addressed. Additionally, we have updated the pdf file to include your suggested revisions and correct any typos.

---

> ### Author Response · Authors · 2024-11-28
>
> Dear Reviewers,
>
>
>
> Thank you once again for your valuable time and feedback. In the spirit of the extended discussion timeline, we have posted a revised version of the submission PDF including the following suggestions, clarifications, and analyses:
>
> Table 9: We have included a comparison of LSTM and single-state ACoE models on continuous action domains to the discrete action domains we had submitted.
>
>
>
> - Tables 10 & 11: We have added ablation studies for hyperparameter sensitivities during training.
>
>
>
> - Section E.3: We have included a detailed discussion of Tables 9, 10, & 11 to the Appendix.
>
>
>
> - Eq 2: We have updated the equations to fix the missing symbols.
>
>
>
> - Line 362: we have revised the section to resolve notational errors.
>
>
>
> We hope the reviewers appreciate the novelty of our formulation and solution to the adversarially perturbed state problem in RL using the partially observable nature of the problem&mdash;we believe this is an impactful development in the area of adversarially robust RL. We hope the reviewers are satisfied with the clarifications made and would be happy to engage in further discussion.

---

### Meta-Review · Area_Chair_nATV · 2024-12-23

**Metareview:**

The paper introduced a new perspective into adversarial RL problems by viewing the state-perturbing adversary as inducing a POMDP. The authors then proposed the concept of Adversarial Counterfactual Error (ACoE) and developed a surrogate of it for applicability in general RL problems. Extensive experiments in various environments demonstrate the effectiveness of the proposed ACoE objective and the practical algorithms.

All reviewers state that this paper makes meaningful contributions to robust RL. The AC agrees and thus recommends acceptance.

**Additional Comments On Reviewer Discussion:**

There were (minor) technical comments, such as the requirement of knowing the attack's radius. The comments were addressed in the rebuttal.

---

### Decision · Program_Chairs · 2025-01-22

Accept (Poster)